# Deep Neuroevolution: Genetic Algorithms are a Competitive Alternative for Training Deep Neural Networks for Reinforcement Learning

## Abstract

Deep artificial neural networks (DNNs) are typically trained via gradient-based learning algorithms, namely backpropagation. Evolution strategies (ES) can rival backprop-based algorithms such as Q-learning and policy gradients on challenging deep reinforcement learning (RL) problems. However, ES can be considered a gradient-based algorithm because it performs stochastic gradient descent via an operation similar to a finite-difference approximation of the gradient. That raises the question of whether non-gradient-based evolutionary algorithms can work at DNN scales. Here we demonstrate they can: we evolve the weights of a DNN with a simple, gradient-free, population-based genetic algorithm (GA) and it performs well on hard deep RL problems, including Atari and humanoid locomotion. The Deep GA successfully evolves networks with over four million free parameters, the largest neural networks ever evolved with a traditional evolutionary algorithm. These results (1) expand our sense of the scale at which GAs can operate, (2) suggest intriguingly that in some cases following the gradient is not the best choice for optimizing performance, and (3) make immediately available the multitude of neuroevolution techniques that improve performance. We demonstrate the latter by showing that combining DNNs with novelty search, which encourages exploration on tasks with deceptive or sparse reward functions, can solve a high-dimensional problem on which reward-maximizing algorithms (e.g. DQN, A3C, ES, and the GA) fail. Additionally, the Deep GA is faster than ES, A3C, and DQN (it can train Atari in ~4 hours on one workstation or ~1 hour distributed on 720 cores), and enables a state-of-the-art, up to 10,000-fold compact encoding technique.

## 1 Introduction

A recent trend in machine learning and AI research is that old algorithms work remarkably well when combined with sufficient computing resources and data. That has been the story for (1) backpropagation applied to deep neural networks in supervised learning tasks such as computer vision Krizhevsky et al. (2012) and voice recognition Seide et al. (2011), (2) backpropagation for deep neural networks combined with traditional reinforcement learning algorithms, such as Q-learning Watkins and Dayan (1992); Mnih et al. (2015) or policy gradient (PG) methods Sehnke et al. (2010); Mnih et al. (2016), and (3) evolution strategies (ES) applied to reinforcement learning benchmarks Salimans et al. (2017). One common theme is that all of these methods are gradient-based, including ES, which involves a gradient approximation similar to finite differences Williams (1992); Wierstra et al. (2008); Salimans et al. (2017). This historical trend raises the question of whether a similar story will play out for gradient-free methods, such as population-based GAs.

This paper investigates that question by testing the performance of a simple GA on hard deep reinforcement learning (RL) benchmarks, including Atari 2600 Bellemare et al. (2013); Brockman et al. (2016); Mnih et al. (2015) and Humanoid Locomotion in the MuJoCo simulator Todorov et al. (2012); Schulman et al. (2015; 2017); Brockman et al. (2016). We compare the performance of the GA with that of contemporary algorithms applied to deep RL (i.e. DQN Mnih et al. (2015), a

Q-learning method, A3C Mnih et al. (2016), a policy gradient method, and ES). One might expect GAs to perform far worse than other methods because they are so simple and do not follow gradients. Surprisingly, we found that GAs turn out to be a competitive algorithm for RL – performing better on some domains and worse on others, and roughly as well overall as A3C, DQN, and ES – adding a new family of algorithms to the toolbox for deep RL problems. We also validate the effectiveness of learning with GAs by comparing their performance to that of random search (RS). While the GA always outperforms random search, interestingly we discovered that in some Atari games random search outperforms powerful deep RL algorithms (DQN on 3/13 games, A3C on 6/13, and ES on 3/13), suggesting that local optima, saddle points, noisy gradient estimates, or other factors are impeding progress on these problems for gradient-based methods. Although deep neural networks often do not struggle with local optima in supervised learning Pascanu et al. (2014), local optima remain an issue in RL because the reward signal may deceptively encourage the agent to perform actions that prevent it from discovering the globally optimal behavior.

Like ES and the deep RL algorithms, the GA has unique benefits. GAs prove slightly faster than ES (discussed below). The GA and ES are thus both substantially faster in wall-clock speed than Q-learning and policy gradient methods. We explore two distinct GA implementations: (1) a single-machine version with GPUs and CPUs, and (2) a distributed version on many CPUs across many machines. On a *single* modern workstation with 4 GPUs and 48 CPU cores, the GA can train Atari in ~4 hours. Training to comparable performance takes ~7-10 days for DQN and ~4 days for A3C. This speedup enables individual researchers with single (albeit expensive) workstations to start using domains formerly reserved for well-funded labs only and iterate perhaps more rapidly than with any other RL algorithm. Given substantial distributed computation (here, 720 CPU cores across dozens of machines), the GA and ES can train Atari in ~1 hour. Also beneficial, via a new technique we introduce, even multi-million-parameter networks trained by GAs can be encoded with very few (thousands of) bytes, yielding the state-of-the-art compact encoding method.

Overall, the unexpectedly competitive performance of the GA (and random search) suggests that the structure of the search space in some of these domains is not amenable to gradient-based search. That realization opens up new research directions on when/how to exploit the regions where a gradient-free search might be more appropriate and motivates research into new kinds of hybrid algorithms.

## 2 BACKGROUND

At a high level, an RL problem challenges an agent to maximize some notion of cumulative reward (e.g. total, or discounted) without supervision as to how to accomplish that goal Sutton and Barto (1998). A host of traditional RL algorithms perform well on small, tabular state spaces Sutton and Barto (1998). However, scaling to high-dimensional problems (e.g. learning to act directly from pixels) was challenging until RL algorithms harnessed the representational power of deep neural networks (DNNs), thus catalyzing the field of deep reinforcement learning (deep RL) Mnih et al. (2015). Three broad families of deep learning algorithms have shown promise on RL problems so far: Q-learning methods such as DQN Mnih et al. (2015), policy gradient methods Sehnke et al. (2010) (e.g. A3C Mnih et al. (2016), TRPO Schulman et al. (2015), PPO Schulman et al. (2017)), and more recently evolution strategies (ES) Salimans et al. (2017).

Deep Q-learning algorithms approximate the optimal Q function with DNNs, yielding policies that, for a given state, choose the action with the maximum Q-value Watkins and Dayan (1992); Mnih et al. (2015); Hessel et al. (2017). Policy gradient methods directly learn the parameters of a DNN policy that outputs the probability of taking each action in each state. A team from OpenAI recently experimented with a simplified version of Natural Evolution Strategies Wierstra et al. (2008), specifically one that learns the mean of a distribution of parameters, but not its variance. They found that this algorithm, which we will refer to simply as evolution strategies (ES), is competitive with DQN and A3C on difficult RL benchmark problems, with much faster training times (i.e. faster wall-clock time when many CPUs are available) due to better parallelization Salimans et al. (2017).

All of these methods can be considered gradient-based methods, as they all calculate or approximate gradients in a DNN and optimize those parameters via stochastic gradient descent/ascent (though they do not require differentiating through the reward function, e.g. a simulator). DQN calculates the gradient of the loss of the DNN Q-value function approximator via backpropagation. Policy gradients sample behaviors stochastically from the current policy and then reinforce those that perform

well via stochastic gradient ascent. ES does not calculate gradients analytically, but approximates the gradient of the reward function in the parameter space Salimans et al. (2017); Wierstra et al. (2008).

Here we test whether a truly gradient-free method, a GA, can perform well on challenging deep RL tasks. We find GAs perform surprisingly well and thus can be considered a new addition to the set of algorithms for deep RL problems.

# 3 METHODS

## 3.1 GENETIC ALGORITHM

We purposefully test with an extremely simple GA to set a baseline for how well evolutionary algorithms work for RL problems. We expect future work to reveal that adding the legion of enhancements that exist for GAs Fogel and Stayton (1994); Haupt and Haupt (2004); Clune et al. (2011); Mouret and Doncieux (2009); Lehman and Stanley (2011a); Stanley et al. (2009); Mouret and Clune (2015) will improve their performance on deep RL tasks.

A genetic algorithm Holland (1992); Eiben et al. (2003) evolves a population $\mathcal{P}$ of $N$ individuals (here, neural network parameter vectors $\theta$, often called *genotypes*). At every *generation*, each $\theta_i$ is evaluated, producing a *fitness* score (aka reward) $F(\theta_i)$. Our GA variant performs *truncation selection*, wherein the top $T$ individuals become the parents of the next generation. To produce the next generation, the following process is repeated $N - 1$ times: A parent is selected uniformly at random with replacement and is *mutated* by applying additive Gaussian noise to the parameter vector: $\theta' = \theta + \sigma\epsilon$ where $\epsilon \sim \mathcal{N}(0, I)$. The appropriate value of $\sigma$ was determined empirically for each experiment, as described in Supplementary Information (SI) Table 2. The $N^{\text{th}}$ individual is an unmodified copy of the best individual from the previous generation, a technique called *elitism*. To more reliably try to select the *true* elite in the presence of noisy evaluation, we evaluate each of the top 10 individuals per generation on 30 additional episodes (counting these frames as ones consumed during training); the one with the highest mean score is the designated elite. Historically, GAs often involve *crossover* (i.e. combining parameters from multiple parents to produce an offspring), but for simplicity we did not include it. The new population is then evaluated and the process repeats for $G$ generations or until some other stopping criterion is met. SI Algorithm 1 provides pseudocode for our version.

Open source code and hyperparameter configurations for all of our experiments are available: `anonymous`. Hyperparameters are also listed in SI Table 2. Hyperparameters were fixed for all Atari games, chosen from a set of 36 hyperparameters tested on six games (Asterix, Enduro, Gravitar, Kangaroo, Seaquest, Venture).

GA implementations traditionally store each individual as a parameter vector $\theta$, but this approach scales poorly in memory and network transmission costs with large populations and large (deeper and wider) neural networks. We propose a novel method to store large parameter vectors compactly by representing each parameter vector as an initialization seed plus the list of random seeds that produced each of the mutations that led to each $\theta$. This information is sufficient to reconstruct each $\theta$. This innovation was critical for an efficient implementation of a *distributed* deep GA. SI Fig. 1 shows, and Eq. 1 describes, the method.

$$\theta^n = \psi(\theta^{n-1}, \tau_n) = \theta^{n-1} + \sigma\varepsilon(\tau_n) \tag{1}$$

where $\theta^n$ is an offspring of $\theta^{n-1}$, $\psi(\theta^{n-1}, \tau_n)$ is a deterministic mutation function, $\tau$ is a vector of mutation seeds that encodes $\theta^n$, $\theta^0 = \phi(\tau_0)$, where $\phi$ is a deterministic initialization function, and $\varepsilon(\tau_n) \sim \mathcal{N}(0, I)$ is a deterministic Gaussian pseudo-random number generator with an input seed $\tau_n$ that produces a vector of length $|\theta|$. In our case, $\varepsilon(\tau_n)$ is a large precomputed table that is indexed by 28-bit seeds. SI Sec. 7.3 provides more details, including how the seeds could be smaller.

This technique is advantageous because the size of the compressed representation increases linearly with the number of generations (often order thousands), and is *independent* of the size of the network (often order millions or more). It does, of course, require computation to reconstruct the DNN weight vector. Competitive Atari-playing agents evolve in as little as tens of generations, enabling a compressed representation of a 4M+ parameter neural network in just thousands of bytes (a 10,000-fold compression). The compression rate depends on the number of generations, but in practice is

always substantial: all Atari final networks were compressible 8,000-50,000-fold. This represents the state of the art in encoding large networks compactly. However, it is not a general network compression technique because it cannot compress arbitrary networks, and instead only works for networks evolved with a GA.

One motivation for choosing ES versus Q-learning and policy gradient methods is its faster wall-clock time with distributed computation, owing to better parallelization Salimans et al. (2017). We found that the distributed CPU-only Deep GA not only preserves this benefit, but slightly improves upon it (SI Sec. 7.1 describes why GAs–distributed or local–are faster than ES). Importantly, GAs can also use GPUs to speed up the forward pass of DNNs (especially large ones), making it possible to train on a single workstation. With our GPU-enabled implementation, on *one* modern workstation we can train Atari in ~4 hours what takes ~1 hour with 720 distributed cores. Distributed GPU training would further speed up training for large population sizes.

## 3.2 NOVELTY SEARCH

One benefit of training deep neural networks with GAs is it enables us to immediately take advantage of algorithms previously developed in the neuroevolution community. As a demonstration, we experiment with novelty search (NS) Lehman and Stanley (2011b), which was designed for deceptive domains in which reward-based optimization mechanisms converge to local optima. NS avoids these local optima by ignoring the reward function during evolution and instead rewarding agents for performing behaviors that have never been performed before (i.e. that are novel). Surprisingly, it can often outperform algorithms that utilize the reward signal, a result demonstrated on maze navigation and simulated biped locomotion tasks Lehman and Stanley (2011b). Here we apply NS to see how it performs when combined with DNNs on a deceptive image-based RL problem (that we call the *Image Hard Maze*). We refer to the GA that optimizes for novelty as GA-NS.

NS requires a behavior characteristic (BC) that describes the behavior of a policy $BC(\pi)$ and a behavioral distance function between the BCs of any two policies: $\text{dist}(BC(\pi_i), BC(\pi_j))$, both of which are domain-specific. After each generation, members of the population have a probability $p$ (here, 0.01) of having their BC stored in an *archive*. The novelty of a policy is defined as the average distance to the $k$ (here, 25) nearest neighbors (sorted by behavioral distance) in the population or archive. Novel individuals are thus determined based on their behavioral distance to current or previously seen individuals. The GA otherwise proceeds as normal, substituting novelty for fitness (reward). For reporting and plotting purposes only, we identify the individual with the highest reward per generation. The algorithm is presented in SI Algorithm 2.

# 4 EXPERIMENTS

Our experiments focus on the performance of the GA on the same challenging problems that have validated the effectiveness of state-of-the-art deep RL algorithms and ES Salimans et al. (2017). They include learning to play Atari directly from pixels Mnih et al. (2015); Schulman et al. (2017); Mnih et al. (2016); Bellemare et al. (2013) and a continuous control problem involving a simulated humanoid robot learning to walk Brockman et al. (2016); Schulman et al. (2017); Salimans et al. (2017); Todorov et al. (2012). We also tested on an Atari-scale maze domain that has a clear local optimum (Image Hard Maze) to study how well these algorithms avoid deception Lehman and Stanley (2011b).

For Atari and Image Hard Maze experiments, we record the best agent found in each of multiple, independent, randomly initialized GA runs: 5 for Atari, 10 for the Image Hard Maze. Because Atari is stochastic, the final score for each run takes the highest-scoring elite across generations, and reports the mean score it achieves on 200 independent evaluations. The final score for the domain is then the median of final run scores. Humanoid Locomotion details are in SI. Sec 7.6.

## 4.1 ATARI

Training deep neural networks to play Atari – mapping directly from pixels to actions – was a celebrated feat that arguably launched the deep RL era and expanded our understanding of the difficulty of RL domains that machine learning could tackle Mnih et al. (2015). Here we test how the perfor-

mance of DNNs evolved by a simple GA compare to DNNs trained by the major families of deep RL algorithms and ES. We model our experiments on those from the ES paper by Salimans et al. (2017) because it inspired our study. Due to limited computational resources, our initial and main study compares results on 13 Atari games. Some were chosen because they are games on which ES performs well (Frostbite, Gravitar, Kangaroo, Venture, Zaxxon) or poorly (Amidar, Enduro, Skiing, Seaquest) and the remaining games were chosen from the ALE Bellemare et al. (2013) set in alphabetical order (Assault, Asterix, Asteroids, Atlantis). We later expanded our study to the full set of 57 Atari games from recent milestone papers Hessel et al. (2017); Horgan et al. (2018) and our conclusions were qualitatively unchanged (SI Sec. 7.8). To facilitate comparisons with results reported in Salimans et al. (2017), we keep the number of game frames agents experience over the course of a GA run constant (at one billion frames). The frame limit results in a differing number of generations per independent GA run (SI Sec. Table 3), as policies of different quality in different runs may see more frames in some games (e.g. if the agent lives longer).

During training, each agent is evaluated on a full episode (capped at 20k frames), which can include multiple lives, and fitness is the sum of episode rewards, i.e. the final Atari game score. The following are identical to DQN Mnih et al. (2015): (1) data preprocessing, (2) network architecture, and (3) the stochastic environment that starts each episode with up to 30 random, initial no-op operations. We use the larger DQN architecture from Mnih et al. (2015) consisting of 3 convolutional layers with 32, 64, and 64 channels followed by a hidden layer with 512 units. The convolutional layers use $8 \times 8$, $4 \times 4$, and $3 \times 3$ filters with strides of 4, 2, and 1, respectively. All hidden layers were followed by a rectifier nonlinearity (ReLU). The network contains over 4M parameters; interestingly, many in the past assumed that a simple GA would fail at such scales. All results are from our single-machine CPU+GPU GA implementation.

Fair comparisons between algorithms is difficult, as evaluation procedures are non-uniform and algorithms realize different trade-offs between computation, wall-clock speed, and sample efficiency. Another consideration is whether agents are evaluated on random starts (a random number of no-op actions), which is the regime they are trained on, or on starts randomly sampled from human play, which tests for generalization Nair et al. (2015). Because we do not have a database of human starts to sample from, our agents are evaluated with random starts. Where possible, we compare our results to those for other algorithms on random starts. That is true for DQN and ES, but not for A3C, where we had to include results on human starts.

We also attempt to control for the number of frames seen during training, but because DQN is far slower to run, we present results from the literature that train on fewer frames (200M, which requires 7-10 days of computation vs. hours of computation needed for ES and the GA to train on 1B frames). There are many variants of DQN that we could compare to, including the Rainbow Hessel et al. (2017) algorithm that combines many different recent improvements to DQN Van Hasselt et al. (2016); Wang et al. (2015); Schaul et al. (2015); Sutton and Barto (1998); Bellemare et al. (2017); Fortunato et al. (2017). However, we choose to compare the GA to the original, vanilla DQN algorithm, partly because we also introduce a vanilla GA, without the many modifications and improvements that have been previously developed Haupt and Haupt (2004).

In what will likely be a surprise to many, the simple GA is able to train deep neural networks to play many Atari games roughly as well as DQN, A3C, and ES (Table 1). Among the first set of 13 games we tried, DQN, ES and the GA produced the best score on 3 games, while A3C produced the best score on 4. On Skiing, the GA produced a score higher than any other algorithm published to date. On some games, the GA performance advantage over DQN, A3C, and ES is considerable (e.g. Frostbite, Venture, Skiing). Videos of policies evolved by the GA can be viewed here: `anonymous`. In a head-to-head comparisons, the GA performs better than ES, A3C, and DQN on 6 games each out of 13 (Tables 1 & 6).

The GA also performs worse on many games, continuing a theme in deep RL where different families of algorithms perform differently across different domains Salimans et al. (2017). However, all such comparisons are preliminary because more computational resources are needed to gather sufficient sample sizes to see if the algorithms are significantly different per game; instead the key takeaway is that they all tend to perform roughly similarly in that each does well on different games.

Because performance did not plateau in the GA runs, we test whether the GA improves further given additional computation. We thus run the GA six times longer (6B frames) and in all games, its score

| | DQN | ES | A3C | RS | GA | GA |
|---|---|---|---|---|---|---|
| Frames | 200M | 1B | 1B | 1B | 1B | 6B |
| Time | ∼7-10d | ∼ 1h | ∼ 4d | ∼ 1h or 4h | ∼ 1h or 4h | ∼ 6h or 24h |
| Forward Passes | 450M | 250M | 250M | 250M | 250M | 1.5B |
| Backward Passes | 400M | 0 | 250M | 0 | 0 | 0 |
| Operations | 1.25B U | 250M U | 1B U | 250M U | 250M U | 1.5B U |
| amidar | **978** | 112 | 264 | 143 | 263 | 377 |
| assault | 4,280 | 1,674 | **5,475** | 649 | 714 | 814 |
| asterix | 4,359 | 1,440 | **22,140** | 1,197 | 1,850 | 2,255 |
| asteroids | 1,365 | 1,562 | **4,475** | 1,307 | 1,661 | 2,700 |
| atlantis | 279,987 | **1,267,410** | 911,091 | 26,371 | 76,273 | 129,167 |
| enduro | **729** | 95 | -82 | 36 | 60 | 80 |
| frostbite | 797 | 370 | 191 | 1,164 | **4,536** | **6,220** |
| gravitar | 473 | **805** | 304 | 431 | 476 | 764 |
| kangaroo | 7,259 | **11,200** | 94 | 1,099 | 3,790 | **11,254** |
| seaquest | **5,861** | 1,390 | 2,355 | 503 | 798 | 850 |
| skiing | -13,062 | -15,443 | -10,911 | -7,679 | **-6,502** | **-5,541** |
| venture | 163 | 760 | 23 | 488 | **969** | **1,422** |
| zaxxon | 5,363 | 6,380 | **24,622** | 2,538 | 6,180 | 7,864 |

Table 1: **On Atari a simple genetic algorithm is competitive with Q-learning (DQN), policy gradients (A3C), and evolution strategies (ES).** Shown are game scores (higher is better). Comparing performance between algorithms is inherently challenging (see main text), but we attempt to facilitate comparisons by showing estimates for the amount of computation (*operations*, the sum of forward and backward neural network passes), data efficiency (the number of game frames from training episodes), and how long in wall-clock time the algorithm takes to run. The ES, DQN, A3C, and GA (1B) perform best on 3, 3, 4, and 3 games, respectively. Thus, overall, each algorithm is best on a different subset of games, and all are in that sense competitive alternatives. The GA produced state-of-the-art results on Skiing. In a much larger set of games, these results qualitatively hold and, surprisingly, the GA can sometimes even outperform highly-sophisticated algorithms produced after years of intense research into improving DQN, such as Rainbow and Ape-X (SI Sec. 7.8). Interestingly, random search often finds policies superior to those of DQN, A3C, and ES (see text for discussion). Note the dramatic differences in the speeds of the algorithm, which are much faster for the GA and ES, and data efficiency, which favors DQN. The scores for DQN are from Hessel et al. (2017) while those for A3C and ES are from Salimans et al. (2017). For A3C, DQN, and ES, we cannot provide error bars because they were not reported in the original literature; GA and random search error bars are visualized in (SI Fig. 2). The wall-clock times are approximate because they depend on a variety of hard-to-control-for factors. We found the GA runs slightly faster than ES on average. GA 6B scores are bolded if best, but do not prevent bolding in other columns.

improves (Table 1). With these post-6B-frame scores, the GA outperforms A3C, ES, and DQN on 7, 8, 7 of the 13 games in head-to-head comparisons, respectively (SI Table 6). In most games, the GA's performance still has not converged at 6B frames (SI Fig. 2), leaving open the question of to how well the GA will ultimately perform when run even longer. To our knowledge, this 4M+ parameter neural network is the largest neural network ever evolved with a simple GA.

In the expanded game set, all of the results described above qualitatively hold. On some games the GA also outperforms Rainbow Hessel et al. (2017) and Ape-X Horgan et al. (2018), two recent, powerful DQN enhancements produced after years of research by a large community into improving DQN (SI Sec. 7.8). The GA yields state-of the-art results on 6 games, including both sparse- and dense-reward games.

One remarkable fact is how quickly the GA finds high-performing individuals. Because we employ a large population size (1K), each run lasts relatively few generations (min 348, max 1,834, SI Table 3). In many games, the GA finds a solution better than DQN in only one or tens of generations! Specifically, the median GA performance is higher than the *final* DQN performance in 1, 1, 3, 5, 11, and 29 generations for Skiing, Venture, Frostbite, Asteroids, Gravitar, and Zaxxon, respectively. Similar results hold for ES, where 1, 2, 3, 7, 12, and 25 GA generations were needed to outperform

ES on Skiing, Frostbite, Amidar, Asterix, Asteroids, and Venture, respectively. The number of generations required to beat A3C were 1, 1, 1, 1, 1, 2, and 52 for Enduro, Frostbite, Kangaroo, Skiing, Venture, Gravitar, and Amidar, respectively.

Each generation, the GA tends to make small-magnitude changes (controlled by $\sigma$) to the parameter vector (see Methods). That the GA outperforms DQN, A3C, and ES in so few generations – especially when it does so in the first generation (which is before a round of selection) – suggests that many high-quality policies exist near the origin (to be precise, in or near the region in which the random initialization function generates policies). That raises the question: is the GA doing anything more than random search?

To answer this question, we evaluate many policies randomly generated by the GA's initialization function $\phi$ and report the best score. We gave random search approximately the same amount of frames and computation as the GA and compared their performance (Table 1). In every game, the GA outperformed random search, and did so significantly on 9/13 games (Fig. 2, $p < 0.05$, this and all future $p$ values are via a Wilcoxon rank-sum test). The improved performance suggests the GA is performing healthy optimization over generations.

Surprisingly, given how celebrated and impressive DQN, ES and A3C are, out of 13 games random search actually outperforms DQN on 3 (Frostbite, Skiing, & Venture), ES on 3 (Amidar, Frostbite, & Skiing), and A3C on 6 (Enduro, Frostbite, Gravitar, Kangaroo, Skiing, & Venture). Interestingly, some of these policies produced by random search are not trivial, degenerate policies. Instead, they appear quite sophisticated. Consider the following example from the game Frostbite, which requires an agent to perform a long sequence of jumps up and down rows of icebergs moving in different directions (while avoiding enemies and optionally collecting food) to build an igloo brick by brick (SI Fig. 3). Only after the igloo is built can the agent enter the igloo to receive a large payoff. Over its first two lives, a policy found by random search completes a series of 17 actions, jumping down 4 rows of icebergs moving in different directions (while avoiding enemies) and back up again three times to construct an igloo. Then, only once the igloo is built, the agent immediately moves towards it and enters it, at which point it gets a large reward. It then repeats the entire process on a harder level, this time also gathering food and thus earning bonus points (video: `anonymous`). That policy resulted in a very high score of 3,620 in less than 1 hour of random search, vs. an average score of 797 produced by DQN after 7-10 days of optimization. One may think that random search found a lucky open loop sequence of actions overfit to that particular stochastic environment. Remarkably, we found that this policy actually generalizes to other initial conditions too, achieving a median score of 3,170 (with 95% bootstrapped median confidence intervals of 2,580 - 3,170) on 200 different test environments (each with up to 30 random initial no-ops, a standard testing procedure Hessel et al. (2017); Mnih et al. (2015)).

These examples and the success of RS versus DQN, A3C, and ES suggest that many Atari games that seem hard based on the low performance of leading deep RL algorithms may not be as hard as we think, and instead that these algorithms for some reason are performing poorly on tasks that are actually quite easy. These results further suggest that sometimes the best search strategy is not to follow the gradient, but instead to conduct a dense search in a local neighborhood and select the best point found, a subject we return to in the discussion (Sec. 5).

## 4.2 IMAGE HARD MAZE

We also conducted an experiment to demonstrate a benefit of GAs working at DNN scales, which is that algorithms that were developed to improve GAs can be immediately taken off the shelf to improve DNN training. The example algorithm we chose is novelty search (NS), a popular evolutionary method for RL exploration Lehman and Stanley (2011b). We found that the GA plus NS can solve a high-dimensional robot control problem on which reward-maximizing algorithms (e.g. DQN, A3C, ES, and the GA) fail (SI Sec. 7.5).

## 4.3 HUMANOID LOCOMOTION

The GA was also able to solve the challenging continuous control benchmark of Humanoid Locomotion Brockman et al. (2016), which has validated modern, powerful algorithms such as A3C, TRPO, and ES. While the GA did produce robots that could walk well, it took ~15 times longer to

perform slightly worse than ES (SI Sec. 7.6), which is surprising because GAs have previously performed well on robot locomotion tasks Clune et al. (2011); Huizinga et al. (2016). Future research is required to understand why.

## 5  DISCUSSION

The surprising success of the GA and RS in domains thought to require at least some degree of gradient estimation suggests some heretofore under-appreciated aspects of high-dimensional search spaces. They imply that densely sampling in a region around the origin is sufficient in some cases to find far better solutions than those found by state-of-the-art, gradient-based methods even with far more computation or wall-clock time, suggesting that gradients do not point to these solutions, or that other optimization issues interfere with finding them, such as saddle points or noisy gradient estimates. The GA results further suggest that sampling in the region around good solutions is often sufficient to find even better solutions, and that a sequence of such discoveries is possible in many challenging domains. That result in turn implies that the distribution of solutions of increasing quality is unexpectedly dense, and that you do not need to follow a gradient to find them.

Another, non-mutually exclusive hypothesis, is that GAs (and ES) have improved performance due to *temporally extended exploration* Osband et al. (2016), meaning they explore consistently because all actions in an episode are a function of the same set of mutated parameters, which improves exploration Plappert et al. (2017). This helps exploration for two reasons: (1) an agent takes the same action (or has the same distribution over actions) each time it visits the same state, which makes it easier to learn whether the policy in that state is advantageous, and (2) the agent is also more likely to have correlated actions across states (e.g. always go up) because mutations to its internal representations can affect the actions taken in many states similarly.

Perhaps more interesting is the result that sometimes it is actually *worse* to follow the gradient than sample locally in the parameter space for better solutions. This scenario probably does not hold in all domains, or even in all the regions of a domain where it sometimes holds, but that it holds at all expands our conceptual understanding of the viability of different kinds of search operators. A reason GA might outperform gradient-based methods is if local optima are present, as it can jump over them in the parameter space, whereas a gradient method cannot (without additional optimization tricks such as momentum, although we note that ES utilized the modern ADAM optimizer in these experiments Kingma and Ba (2014), which includes momentum). One unknown question is whether GA-style local, gradient-free search is better early on in the search process, but switching to a gradient-based search later allows further progress that would be impossible, or prohibitively computationally expensive, for a GA to make. Another unknown question is the promise of simultaneously hybridizing GA methods with modern algorithms for deep RL, such as Q-learning, policy gradients, or evolution strategies.

We still know very little about the ultimate promise of GAs versus competing algorithms for training deep neural networks on reinforcement learning problems. Additionally, here we used an extremely simple GA, but many techniques have been invented to improve GA performance Eiben et al. (2003); Haupt and Haupt (2004), including crossover Holland (1992); Deb and Myburgh (2016), indirect encoding Stanley (2007); Stanley et al. (2009); Clune et al. (2011), and encouraging quality diversity Mouret and Clune (2015); Pugh et al. (2016), just to name a few. Moreover, many techniques have been invented that dramatically improve the training of DNNs with backpropagation, such as residual networks He et al. (2015), SELU or RELU activation functions Krizhevsky et al. (2012); Klambauer et al. (2017), LSTMs or GRUs Hochreiter and Schmidhuber (1997); Cho et al. (2014), regularization Hoerl and Kennard (1970), dropout Srivastava et al. (2014), and annealing learning rate schedules Robbins and Monro (1951). We hypothesize that many of these techniques will also improve neuroevolution for large DNNs.

Some of these enhancements may improve the GA performance on Humanoid Locomotion. For example, indirect encoding, which allows genomic parameters to affect multiple weights in the final neural network (in a way similar to convolution's tied weights, but with far more flexibility), has been shown to dramatically improve performance and data efficiency when evolving robot gaits Clune et al. (2011). Those results were found with the HyperNEAT algorithm Stanley et al. (2009), which has an indirect encoding that abstracts the power of developmental biology Stanley (2007), and is a particularly promising direction for Humanoid Locomotion and Atari we are investigating.

It will further be interesting to learn on which domains Deep GA tends to perform well or poorly and understand why. Also, GAs could help in other non-differentiable domains, such as architecture search Liu et al. (2017); Miikkulainen et al. (2017) and for training limited precision (e.g. binary) neural networks.

## 6 CONCLUSION

Our work introduces a Deep GA that competitively trains deep neural networks for challenging RL tasks, and an encoding technique that enables efficient distributed training and a state-of-the-art compact network encoding. We found that the GA is fast, enabling training Atari in ~4h on a single workstation or ~1h distributed on 720 CPUs. We documented that GAs are surprisingly competitive with popular algorithms for deep reinforcement learning problems, such as DQN, A3C, and ES, especially in the challenging Atari domain. We also showed that interesting algorithms developed in the neuroevolution community can now immediately be tested with deep neural networks, by showing that a Deep GA-powered novelty search can solve a deceptive Atari-scale game. It will be interesting to see future research investigate the potential and limits of GAs, especially when combined with other techniques known to improve GA performance. More generally, our results continue the story – started by backprop and extended with ES – that old, simple algorithms plus modern amounts of computation can perform amazingly well. That raises the question of what other classic algorithms should be revisited.

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

# 7 SUPPLEMENTARY INFORMATION

## 7.1 WHY THE GA IS FASTER THAN ES

The GA is faster than ES for two main reasons: (1) for every generation, ES must calculate how to update its neural network parameter vector $\theta$. It does so via a weighted average across many (10,000 in Salimans et al. (2017)) *pseudo-offspring* (random $\theta$ perturbations) weighted by their fitness. This averaging operation is slow for large neural networks and large numbers of pseudo-offspring (the latter is required for healthy optimization), and is not required for the Deep GA. (2) ES requires virtual batch normalization to generate diverse policies amongst the pseudo-offspring, which is necessary for accurate finite difference approximation Salimans et al. (2016). Virtual batch normalization requires additional forward passes for a reference batch–a random set of observations chosen at the start of training–to compute layer normalization statistics that are then used in the same manner as batch normalization Ioffe and Szegedy (2015). We found that the random GA parameter perturbations generate sufficiently diverse policies without virtual batch normalization and thus avoid these additional forward passes through the network.

---

**Algorithm 1** Simple Genetic Algorithm

---

**Input:** mutation function $\psi$, population size $N$, number of selected individuals $T$, policy initialization routine $\phi$, fitness function $F$.
**for** $g = 1, 2..., G$ generations **do**
    **for** $i = 1, ..., N - 1$ in next generation's population **do**
        **if** $g = 1$ **then**
            $\mathcal{P}_i^{g=1} = \phi(\mathcal{N}(0, I))$ {initialize random DNN}
        **else**
            $k = \text{uniformRandom}(1, T)$ {select parent}
            $\mathcal{P}_i^g = \psi(\mathcal{P}_k^{g-1})$ {mutate parent}
        **end if**
        Evaluate $F_i = F(\mathcal{P}_i^g)$
    **end for**
    Sort $\mathcal{P}_i^g$ with descending order by $F_i$
    **if** $g = 1$ **then**
        Set Elite Candidates $C \leftarrow \mathcal{P}_{1...10}^{g=1}$
    **else**
        Set Elite Candidates $C \leftarrow \mathcal{P}_{1...9}^g \cup \{\text{Elite}\}$
    **end if**
    Set Elite $\leftarrow \arg\max_{\theta \in C} \frac{1}{30} \sum_{j=1}^{30} F(\theta)$
    $\mathcal{P}^g \leftarrow [\text{Elite}, \mathcal{P}^g - \{\text{Elite}\}]$ {only include elite once}
**end for**
**Return: Elite**

---

## 7.2 HYPERPARAMETERS

We use Xavier initialization Glorot and Bengio (2010) as our policy initialization function $\phi$ where all bias weights are set to zero, and connection weights are drawn from a standard normal distribution with variance $1/N_{in}$, where $N_{in}$ is the number of incoming connections to a neuron.

## 7.3 ADDITIONAL INFORMATION ABOUT THE DEEP GA COMPACT ENCODING METHOD

The compact encoding technique is based on the principle that the seeds need only be long enough to generate a unique mutation vector per offspring per parent. If any given parent $\theta^{n-1}$ produces at most $x$ offspring, then $\tau_n$ in Eq. 1 can be as small as a $log_2(x)$-bit number. $\tau_0$ is a special case that needs one unique seed for each of the $N$ $\theta$ vectors in generation 0, and can thus be encoded with $log_2(N)$ bits. The reason the seed bit-length can be vastly smaller than the search space size is because not every point in the search space is a possible offspring of $\theta^n$, and we only need to be able generate offspring randomly (we do not need to be able to reach any point in the search space

---

**Algorithm 2** Novelty Search (GA-NS)

---

**Input:** mutation function $\psi$, population size $N$, number of selected individuals $T$, policy initialization routine $\phi$, empty archive $\mathcal{A}$, archive insertion probability $p$, a novelty function $\eta$, a behavior characteristic function $BC$.
**for** $g = 1, 2..., G$ generations **do**
    **for** $i = 2, ..., N$ in next generation's population **do**
      **if** $g = 1$ **then**
        $\mathcal{P}_i^{g=1} = \phi(\mathcal{N}(0, I))$ {initialize random DNN}
      **else**
        $k = \text{uniformRandom}(1, T)$ {select parent}
        $\mathcal{P}_i^g = \psi(\mathcal{P}_k^{g-1})$ {mutate parent}
      **end if**
      $BC_i^g = BC(\mathcal{P}_i^g)$
    **end for**
    Copy $\mathcal{P}_1^g \leftarrow \mathcal{P}_1^{g-1}$; $BC_1^g \leftarrow BC_1^{g-1}$
    **for** $i = 1, ..., N$ in next generation's population **do**
      Evaluate $F_i = \eta(BC_i^g, (\mathcal{A} \cup BC^g) - \{BC_i^g\})$
      **if** $i > 1$ **then**
        Add $BC_i^g$ to $\mathcal{A}$ with probability $p$
      **end if**
    **end for**
    Sort $\mathcal{P}_i^g$ with descending order by $F_i$
**end for**
**Return: Elite**

---

| Hyperparameter | Humanoid Locomotion | Image Hard Maze | Atari |
|---|---|---|---|
| Population Size (N) | 12,500+1 | 20,000+1 | 1,000+1 |
| Mutation Power ($\sigma$) | 0.00224 | 0.005 | 0.002 |
| Truncation Size (T) | 625 | 61 | 20 |
| Number of Trials | 5 | 1 | 1 |
| Archive Probability | | 0.01 | |

Table 2: **Hyperparameters.** Population sizes are incremented to account for elites ($+1$). Many of the unusual numbers were found via preliminary hyperparameter searches in other domains.

in one random step). However, because we perform $n$ random mutations to produce $\theta^n$, the process can reach many points in the search space.

However, to truly be able to reach every point in the search space we need our set of mutation vectors to span the search space, meaning we need the seed to be at least $log_2(|\theta|)$ bits. To do so we can use a function $\mathcal{H}(\theta, \tau)$ that maps a given $(\theta, \tau_n)$-pair to a new seed and applies it as such:

$$\psi(\theta^{n-1}, \tau_n) = \theta^{n-1} + \varepsilon(\mathcal{H}(\theta^{n-1}, \tau_n)) \tag{2}$$

Note that in this case there are two notions of seeds. The encoding is a series of small $\tau$ seeds, but each new seed $\tau$ is generated from the parent $\theta$ and the previous seed.

### 7.4 ADDITIONAL EXPERIMENTAL DETAILS FOR THE IMAGE HARD MAZE

For temporal context, the current frame and previous three frames are all input at each timestep, following Mnih et al. (2015). The outputs remain the same as in the original Hard Maze problem formulation in Lehman and Stanley (2011b). Unlike the Atari domain, the Image Hard Maze environment is deterministic and does not need multiple evaluations of the same policy.

Following Lehman and Stanley (2011b), the BC is the $(x, y)$ position of the robot at the end of the episode (400 timesteps), and the behavioral distance function is the squared Euclidean distance

| Game | Minimum Generations | Median Generations | Maximum Generations |
|---|---|---|---|
| amidar | 1325 | 1364 | 1541 |
| assault | 501 | 707 | 1056 |
| asterix | 494 | 522 | 667 |
| asteroids | 1096 | 1209 | 1261 |
| atlantis | 507 | 560 | 580 |
| enduro | 348 | 348 | 348 |
| frostbite | 889 | 1016 | 1154 |
| gravitar | 1706 | 1755 | 1834 |
| kangaroo | 688 | 787 | 862 |
| seaquest | 660 | 678 | 714 |
| skiing | 933 | 1237 | 1281 |
| venture | 527 | 606 | 680 |
| zaxxon | 765 | 810 | 823 |

Table 3: **The number of generations at which the GA reached 6B frames.**

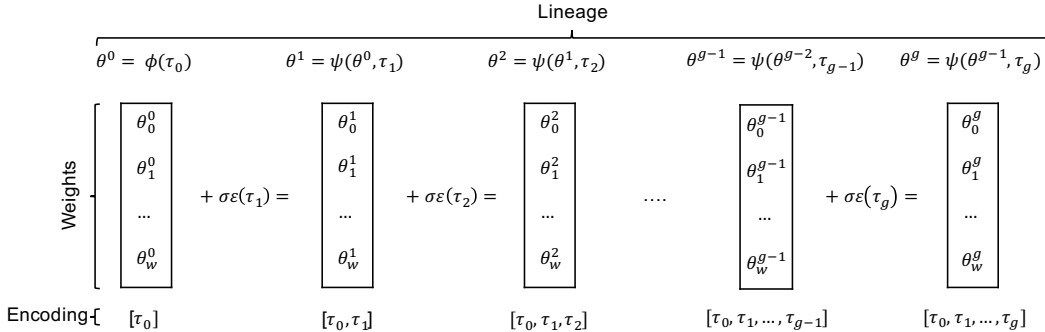

Figure 1: **Visual representation of the Deep GA encoding method.** From a randomly initialized parameter vector $\theta^0$ (produced by an initialization function $\phi$ seeded by $\tau_0$), the mutation function $\psi$ (seeded by $\tau_1$) applies a mutation that results in $\theta^1$. The final parameter vector $\theta^g$ is the result of a series of such mutations. Recreating $\theta^g$ can be done by applying the mutation steps in the same order. Thus, knowing the series of seeds $\tau_0...\tau_g$ that produced this series of mutations is enough information to reconstruct $\theta^g$ (the initialization and mutation functions are deterministic). Since each $\tau$ is small (here, 28 bits long), and the number of generations is low (order hundreds or thousands), a large neural network parameter vector can be stored compactly.

between these final $(x, y)$ positions. The simulator ignores forward or backward motion that would result in the robot penetrating walls, preventing a robot from sliding along a wall, although rotational motor commands still have their usual effect in such situations.

## 7.5 IMAGE HARD MAZE

This experiment seeks to demonstrate a benefit of GAs working at DNN scales, which is that algorithms that were developed to improve GAs can be immediately taken off the shelf to improve DNN training. The example algorithm is novelty search (NS), which is a popular evolutionary method for exploration in RL Lehman and Stanley (2011b).

NS was originally motivated by the Hard Maze domain Lehman and Stanley (2011b), which is a staple in the neuroevolution community. It demonstrates the problem of local optima (aka deception) in reinforcement learning. In it, a robot receives more reward the closer it gets to the goal as the crow flies. The problem is deceptive because greedily getting closer to the goal leads an agent to permanently get stuck in one of the map's deceptive traps (Fig. 5, Left). Optimization algorithms

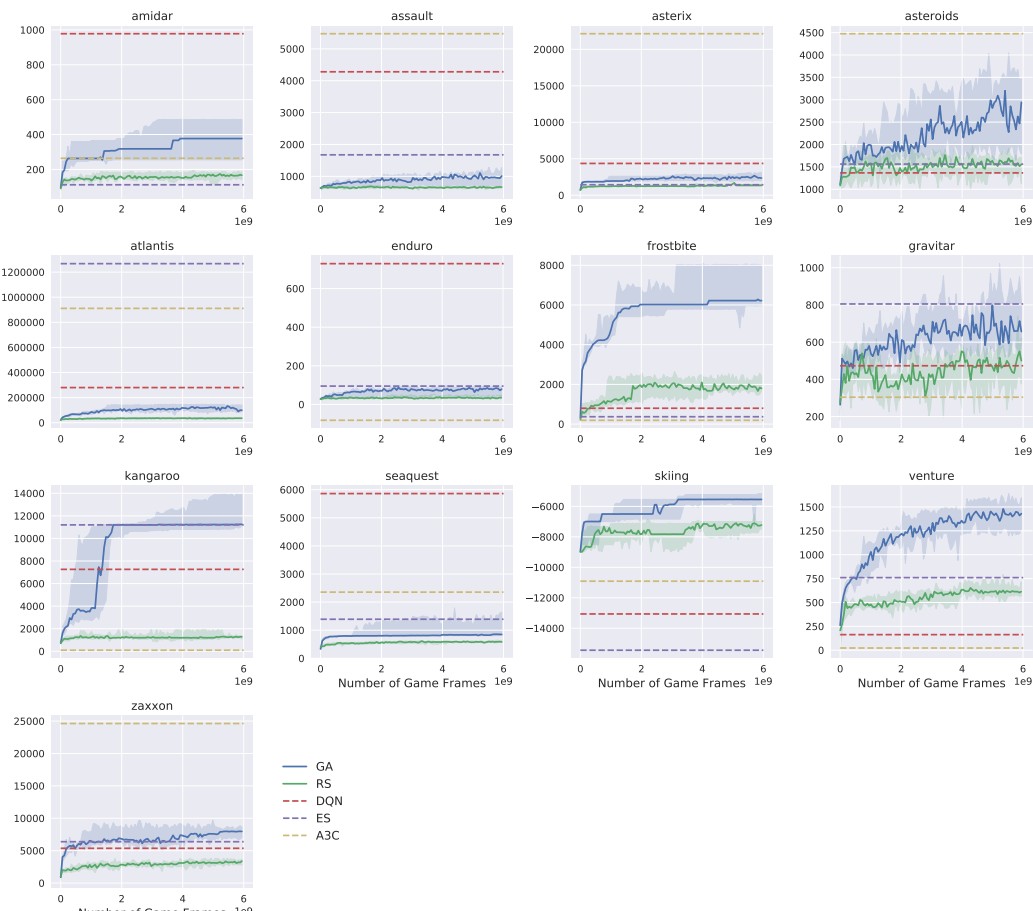

Figure 2: **GA and random search performance across generations on Atari 2600 games.** The performance of the GA and random search compared to DQN, A3C, and ES depends on the game. We plot final scores (as dashed lines) for DQN, A3C, and ES because we do not have their performance values across training and because they trained on different numbers of game frames (SI Table 1). For GA and RS, we report the median and 95% bootstrapped confidence intervals of the median across 5 experiments of the current elite per run, where the score for each elite is a mean of 30 independent episodes.

that do not conduct sufficient exploration suffer this fate. NS solves this problem because it ignores the reward and encourages agents to visit new places Lehman and Stanley (2011b).

The original version of this problem involves only a few inputs (radar sensors to sense walls) and two continuous outputs for speed (forward or backward) and rotation, making it solvable by small neural networks (tens of connections). Because here we want to demonstrate the benefits of NS at the scale of deep neural networks, we introduce a new version of the domain called Image Hard Maze. Like many Atari games, it shows a bird's-eye view of the world to the agent in the form of an $84 \times 84$ pixel image (Fig. 5, Left). This change makes the problem easier in some ways (e.g. now it is fully observable), but harder in others because it is much higher-dimensional: the neural network must learn to process this pixel input and take actions. SI Sec. 7.4 has additional experimental details.

We confirm that the results that held for small neural networks on the original, radar-based version of this task also hold for the high-dimensional, visual version of this task with deep neural networks. With a 4M+ parameter network processing pixels, the GA-based novelty search (GA-NS) is able to solve the task by finding the goal (Fig. 5). The GA optimizes for reward only and, as expected,

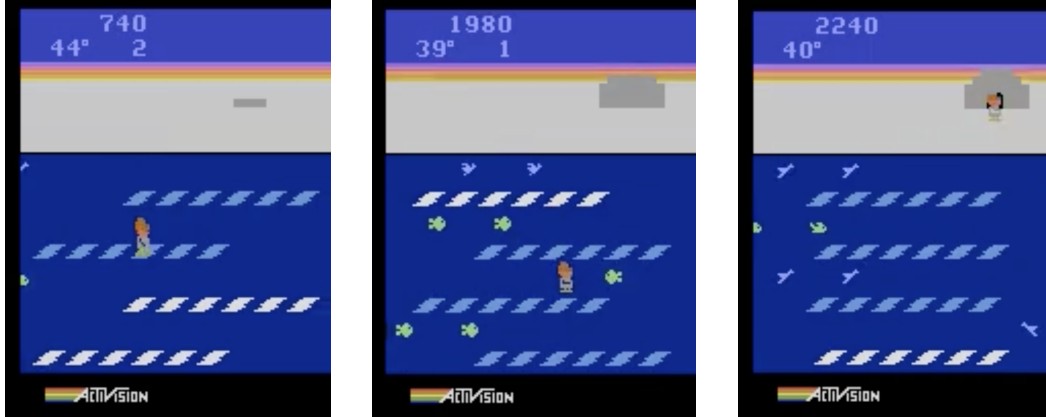

Figure 3: **Example of high-performing individual on Frostbite found through random search**. See main text for a description of the behavior of this policy. Its final score is 3,620 in this episode, which is far higher than the scores produced by DQN, A3C and ES, although not as high as the score found by the GA (Table 1).

gets stuck in the local optima of Trap 2 (SI Fig. 4) and thus fails to solve the problem (Fig. 5), significantly underperforming GA-NS ($p < 0.001$). Our results confirm that we are able to use exploration methods such as novelty search to solve this sort of deception, even in high-dimensional problems such as those involving learning directly from pixels. This is the largest neural network optimized by novelty search to date by three orders of magnitude. In a paper published concurrently with ours, Conti et al. (2017) demonstrate a similar finding, by hybridizing novelty search with ES to create NS-ES, and show that it too can help deep neural networks avoid deception in challenging RL benchmark domains.

As expected, ES also fails to solve the task because it focuses solely on maximizing reward (Fig. 5 & SI Fig. 4). We also test Q-learning (DQN) and policy gradients on this problem. We did not have source code for A3C, but were able to obtain source code for A2C, which has similar performance Wu et al. (2017): the only difference is that it is synchronous instead of asynchronous. For these experiments we modified the rewards of the domain to step-by-step rewards (the negative change in distance to goal since the last time-step), but for plotting purposes, we record the final distance to the goal. Having per-step rewards is standard for these algorithms and provides more information, but does not remove the deception. Because DQN requires discrete outputs, for it we discretize each of the two continuous outputs into to five equally sized bins. To enable all possible output combinations, it learns $5^2 = 25$ Q-values.

Also as expected, DQN and A2C fail to solve this problem (Fig. 5, SI Fig. 4). Their default exploration mechanisms are not enough to find the global optimum given the deceptive reward function in this domain. DQN is drawn into the expected Trap 2. For unclear reasons, even though A2C visits Trap 2 often early in training, it converges on getting stuck in a different part of the maze. Of course, exploration techniques could be added to these controls to potentially make them perform as well as GA-NS. Here we only sought to show that the Deep GA allows algorithms developed for small-scale neural networks can be harnessed on hard, high-dimensional problems that require DNNs.

In future work, it will be interesting to combine NS with a Deep GA on more domains, including Atari and robotics domains. More importantly, our demonstration suggests that other algorithms that enhance GAs can now be combined with DNNs. Perhaps most promising are those that combine a notion of diversity (e.g. novelty) and quality (i.e. being high performing), seeking to collect a set of high-performing, yet interestingly different policies Mouret and Clune (2015); Lehman and Stanley (2011a); Cully et al. (2015); Pugh et al. (2016). The results also motivate future research into combining deep RL algorithms (e.g. DQN, A3C) with novelty search and quality diversity algorithms.

## 7.6 HUMANOID LOCOMOTION

We tested the GA on a challenging continuous control problem, specifically humanoid locomotion. We test with the MuJoCo Humanoid-v1 environment in OpenAI Gym Todorov et al. (2012); Brockman et al. (2016), which involves a simulated humanoid robot learning to walk. Solving this problem has validated modern, powerful algorithms such as A3C Mnih et al. (2016), TRPO Schulman et al. (2015), and ES Salimans et al. (2017).

This problem involves mapping a vector of 376 scalars that describe the state of the humanoid (e.g. its position, velocity, angle) to 17 joint torques. The robot receives a scalar reward that is a combination of four components each timestep. It gets positive reward for standing and its velocity in the positive $x$ direction, and negative reward the more energy it expends and for how hard it impacts the ground. These four terms are summed over every timestep in an episode to calculate the total reward.

To stabilize training, we normalize each dimension of the input by subtracting its mean and dividing by its standard deviation, which are computed from executing 10,000 random policies in the environment. We also applied annealing to the mutation power $\sigma$, decreasing it to 0.001 after 1,000 generations, which resulted in a small performance boost at the end of training. The full set of hyperparameters are listed in SI Table 2.

For these experiments we ran 5 independent, randomly initialized, runs and report the median of those runs. During the elite selection routine we did not reevaluate offspring 30 times like on the Atari experiments. That is because we ran these experiments before the Atari experiments, and we improved our evaluation methods after these experiments were completed. We did not have the computational resources to re-run these experiments with the changed protocol, but we do not believe this change would qualitatively alter our results. We also used the normalized columns initialization routine of Salimans et al. (2017) instead of Xavier initialization, but we found them to perform qualitatively similarly. When determining the fitness of each agent we evaluate the mean over 5 independent episodes. After each generation, for plotting purposes only, we evaluate the elite 30 times.

The architecture has two 256-unit hidden layers with tanh activation functions. This architecture is the one in the configuration file included in the source code released by Salimans et al. (2017). The architecture described in their paper is similar, but smaller, having 64 neurons per layer Salimans et al. (2017). Although relatively shallow by deep learning standards, and much smaller than the Atari DNNs, this architecture still contains ∼167k parameters, which is orders of magnitude greater than the largest neural networks evolved for robotics tasks that we are aware of, which contained 1,560 Huizinga et al. (2016) and before that 800 parameters Clune et al. (2011). Many assumed evolution would fail at larger scales (e.g. networks with hundreds of thousands or millions of weights, as in this paper).

Previous work has called the Humanoid-v1 problem solved with a score of ∼6,000 Salimans et al. (2017). The GA achieves a median above that level after ∼1,500 generations. However, it requires far more computation than ES to do so (ES requires ∼100 generations for median performance to surpass the 6,000 threshold). It is not clear why the GA requires so much more computation, especially given how quickly the GA found high-performing policies in the Atari domain. It is also surprising that the GA does not excel at this domain, given that GAs have performed well in the past on robot control tasks Clune et al. (2011). While the GA needs far more computation in this domain, it is interesting nevertheless that it does eventually solve it by producing an agent that can walk and score over 6,000. Considering its very fast discovery of high-performing solutions in Atari, clearly the GA's advantage versus other methods depends on the domain, and understanding this dependence is an important target for future research.

## 7.7 THE MEANING OF "FRAMES"

Many papers, including ours, report the number of "frames" used during training. However, it is a bit unclear in the literature what is meant exactly by this term. We hope to introduce some terminology that can lend clarity to this confusing issue, which will improve reproducibility and our ability to compare algorithms fairly. Imagine if the Atari-emulator emitted 4B frames during training. We suggest calling these "game frames." One could sub-sample every 4th frame (indeed, due to "frame

|        | DQN | ES | A3C | RS 1B | GA 1B | GA 6B |
|--------|-----|----|-----|-------|-------|-------|
| DQN    |     | 6  | 6   | 3     | 6     | 7     |
| ES     | 7   |    | 7   | 3     | 6     | 8     |
| A3C    | 7   | 6  |     | 6     | 6     | 7     |
| RS 1B  | 10  | 10 | 7   |       | 13    | 13    |
| GA 1B  | 7   | 7  | 7   | 0     |       | 13    |
| GA 6B  | 6   | 5  | 6   | 0     | 0     |       |

Table 4: **Head-to-head comparison between algorithms on the 13 Atari games.** Each value represents how many games for which the algorithm listed at the top of a column produces a higher score than the algorithm listed to the left of that row (e.g. GA 6B beats DQN on 7 games).

skip", most Atari papers do exactly this, and repeat the previous action for each skipped frame), resulting in 1B frames. We suggest calling these 1B frames "training frames", as these are the frames the algorithm is trained on. In our paper we report the *game frames* used by each algorithm. Via personal communication with scientists at OpenAI and DeepMind, we confirmed that we are accurately reporting the number of frames (and that they are game frames, not training frames) used by DQN, A3C, and ES in Mnih et al. (2015), Mnih et al. (2016), and Salimans et al. (2017), respectively.

There is one additional clarification. In all of the papers just mentioned and for the GA in this paper, the input to the network for Atari is the current frame$_t$ and three previous frames. These three previous frames are from the *training frame set*, meaning that if $t$ counts each *game frame* then the input to the network is the following: game frame$_t$, frame$_{t-4}$, frame$_{t-8}$, and frame$_{t-12}$.

### 7.8 Experiments on an expanded set of Atari games and comparisons against modern, powerful DQN variants (Rainbow and Ape-X)

To test whether our results hold on a larger set of Atari games, we extended our experiments to the full 57-game set of Atari games used in two recent, famous papers that described enhancements to DQN, each of which improved the state of the art when published: Rainbow Hessel et al. (2017) and an algorithm that came out after our paper was published on arXiv, Ape-X Horgan et al. (2018). Two of the games did not run due to a bug in OpenAI's Gym Brockman et al. (2016), leaving us with 55 total games. This set is a superset of the games from Mnih et al. (2015). We also added performance comparisons to two strong, recent DQN variants. For these experiments, the score within each run was a median over many (200) evaluations of the policy, instead of the mean (as done for our original 13 games), which we switched to because it is more robust to outliers: doing so lowers the scores somewhat because extreme outliers tend to be very high scores. The results can be seen in Table 5 and head-to-head tallies are in Table 6.

On this much larger set of games, the overall qualitative conclusions of our paper remain unchanged: the simple GA is roughly even with simple RL algorithms such as A3C and DQN, and ES, in that all of these algorithms can learn to play many Atari games well, and each is best on a subset of the games. The GA, which most assumed would not work at all at optimizing large, deep, multi-million parameter neural networks, especially in comparison to DQN and A3C, achieves superior performance to DQN and A3C on 43% (22 of 51, tying on 1) and 49% (27 of 55) games, respectively. Additionally, the GA achieves state of the art results on 6 games (bowling, centipede, private_eye, skiing, solaris). The GA also also exhibits super-human performance on 43% of games (and is the only algorithm we are aware of with super-human performance on bowling). Each game is idiosyncratic, and could be considered a separate domain. Because the GA is roughly as good as some of the most famous Deep RL algorithms (DQN, A3C, and ES), and far better on some games, it is a valuable additional tool to have in our toolbox. When compared against Rainbow Hessel et al. (2017), which combines many of the best innovations built on top of DQN over years by a large research community, the GA still performs better on some games (it wins 11 of 52, ties on 2, and loses on 39: Table 6). The same is true when comparing against the Ape-X algorithm Horgan et al. (2018), which is a very recent, powerful version of prioritized DQN Schaul et al. (2015) (prioritized DQN is already an important improvement over the simple DQN) with a large number of distributed data-gathering agents each running epsilon-greedy exploration with different epsilons: the large

amount of data generated and the different epsilons help with exploration and learning. Ape-X still underperforms the GA on 7 games. As discussed at more length in the main text, the Deep GA thus provides an interesting alternative algorithm to have added to the toolbox for deep RL problems: It may be practically helpful on any given domain, raises interesting new research questions into why it succeeds where other algorithms fail (and vice versa), and because we tested such a simple GA it is still unknown how its performance will compare to other algorithms once enhancements known to improve GA performance Fogel and Stayton (1994); Haupt and Haupt (2004); Clune et al. (2011); Mouret and Doncieux (2009); Lehman and Stanley (2011a); Stanley et al. (2009); Mouret and Clune (2015) are added to it.

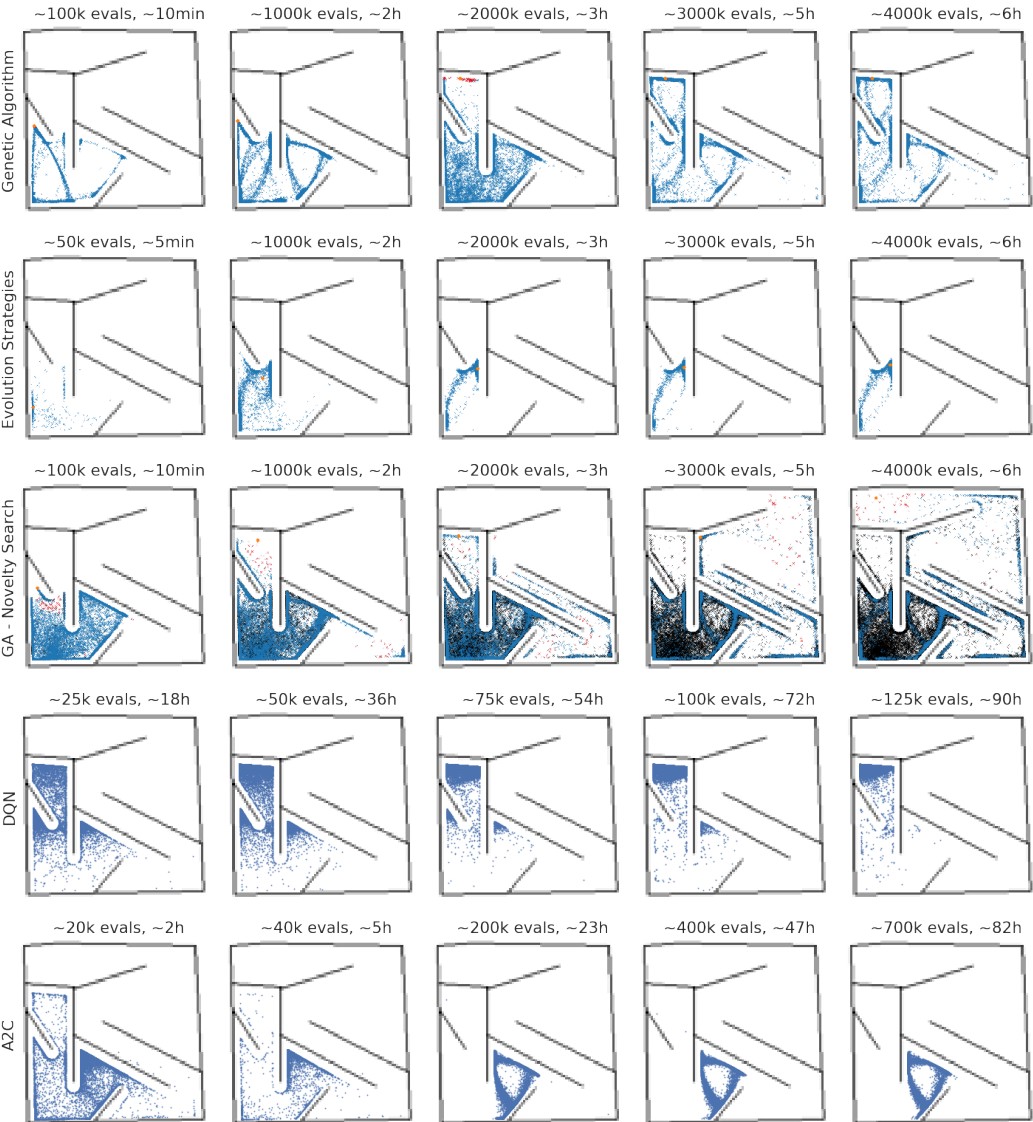

Figure 4: **How different algorithms explore the deceptive Image Hard Maze over time.** Traditional reward-maximization algorithms do not exhibit sufficient exploration to avoid the local optimum (of going up into Trap 2, as shown in Fig. 5). In contrast, a GA optimizing for novelty only (GA-NS) explores the entire environment and ultimately finds the goal. For the evolutionary algorithms (GA-NS, GA, ES), blue crosses represent the population (pseudo-offspring for ES), red crosses represent the top $T$ GA offspring, orange dots represent the final positions of GA elites and the current mean ES policy, and the black crosses are entries in the GA-NS archive. All 3 evolutionary algorithms had the same number of evaluations, but ES and the GA have many overlapping points because they revisit locations due to poor exploration, giving the illusion of fewer evaluations. For DQN and A2C, we plot the end-of-episode position of the agent for each of the 20K episodes prior to the checkpoint listed above the plot. It is surprising that ES significantly underperforms the GA ($p < 0.001$). In 8 of 10 runs it gets stuck near Trap 1, not because of deception, but instead seemingly because it cannot reliably learn to pass through a small bottleneck corridor. This phenomenon has never been observed with population-based GAs on the Hard Maze, suggesting the ES (at least with these hyperparameters) is qualitatively different than GAs in this regard Lehman et al. (2017). We believe this difference occurs because ES optimizes for the average reward of the population sampled from a probability distribution. Even if the maximum fitness of agents sampled from that distribution is higher further along a corridor, ES will not move in that direction if the population average is lower (e.g. if other policies sampled from the distribution crash into the walls, or experience other low-reward fates) Lehman et al. (2017). Note, however, that even when ES moved through this bottleneck (2 out of 10 runs), because it is solely reward-driven, it got stuck in Trap 2.

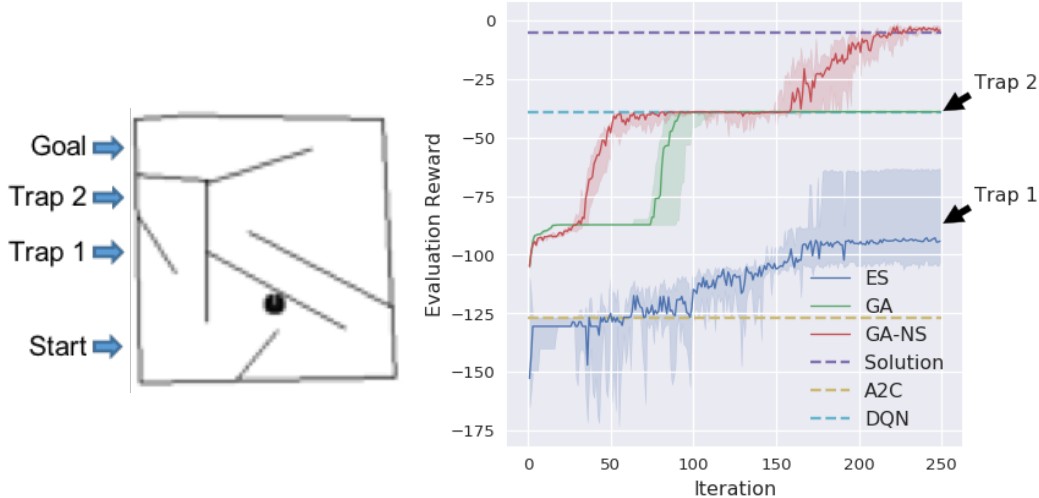

Figure 5: **Image Hard Maze Domain and Results.** Left: A small wheeled robot must navigate to the goal with this bird's-eye view as pixel inputs. The robot starts in the bottom left corner facing right. Right: novelty search can train deep neural networks to avoid local optima that stymie other algorithms. The GA, which solely optimizes for reward and has no incentive to explore, gets stuck on the local optimum of Trap 2. The GA optimizing for novelty (GA-NS) is encouraged to ignore reward and explore the whole map, enabling it to eventually find the goal. ES performs even worse than the GA, as discussed in the main text. DQN and A2C also fail to solve this task. For ES, the performance of the mean $\theta$ policy each iteration is plotted. For GA and GA-NS, the performance of the highest-scoring individual per generation is plotted. Because DQN and A2C do not have the same number of evaluations per iteration as the evolutionary algorithms, we plot their final median reward as dashed lines. SI Fig. 4 shows the behavior of these algorithms during training.

| | Human | DQN | Rainbow | Ape-X | A3C | ES | GA 6B |
|---|---|---|---|---|---|---|---|
| alien | 6,875.0 | 1,620.0 | 9,491.7 | **40,804.9** | 518.4 | | 1,990.0 |
| amidar | 1,676.0 | 978.0 | 5,131.2 | **8,659.2** | 263.9 | 112.0 | 370.0 |
| assault | 1,496.0 | 4,280.0 | 14,198.5 | **24,559.4** | 5,474.9 | 1,673.9 | 898.0 |
| asterix | 8,503.0 | 4,359.0 | **428,200.3** | 313,305.0 | 22,140.5 | 1,440.0 | 1,800.0 |
| asteroids | 13,157.0 | 1,364.5 | 2,712.8 | **155,495.1** | 4,474.5 | 1,562.0 | 1,940.0 |
| atlantis | 29,028.0 | 279,987.0 | 826,659.5 | 944,497.5 | 911,091.0 | **1,267,410.0** | 57,300.0 |
| bank_heist | 734.4 | 455.0 | 1,358.0 | **1,716.4** | 970.1 | 225.0 | 270.0 |
| battle_zone | 37,800.0 | 29,900.0 | 62,010.0 | **98,895.0** | 12,950.0 | 16,600.0 | 25,000.0 |
| beam_rider | 5,775.0 | 8,627.5 | 16,850.2 | **63,305.2** | 22,707.9 | 744.0 | 756.0 |
| berzerk | | 585.6 | 2,545.6 | **57,196.7** | 817.9 | 686.0 | 1,440.0 |
| bowling | 154.8 | 50.4 | 30.0 | 17.6 | 35.1 | 30.0 | **197.0** |
| boxing | 4.3 | 88.0 | 99.6 | **100.0** | 59.8 | 49.8 | 64.0 |
| breakout | 31.8 | 385.5 | 417.5 | **800.9** | 681.9 | 9.5 | 10.0 |
| centipede | 11,963.0 | 4,657.7 | 8,167.3 | 12,974.0 | 3,755.8 | 7,783.9 | **14,122.0** |
| chopper_command | 9,882.0 | 6,126.0 | 16,654.0 | **721,851.0** | 7,021.0 | 3,710.0 | 3,500.0 |
| crazy_climber | 35,411.0 | 110,763.0 | 168,788.5 | **320,426.0** | 112,646.0 | 26,430.0 | 38,000.0 |
| demon_attack | 3,401.0 | 12,149.4 | 111,185.2 | **133,086.4** | 113,308.4 | 1,166.5 | 970.0 |
| double_dunk | -15.5 | -6.6 | -0.3 | **23.5** | -0.1 | 0.2 | 0.0 |
| enduro | 309.6 | 729.0 | 2,125.9 | **2,177.4** | -82.5 | 95.0 | 51.0 |
| fishing_derby | 5.5 | -4.9 | 31.3 | 44.4 | 18.8 | **49.0** | -33.0 |
| freeway | 29.6 | 30.8 | **34.0** | 33.7 | 0.1 | 31.0 | 26.0 |
| frostbite | 4,335.0 | 797.4 | **9,590.5** | 9,328.6 | 190.5 | 370.0 | 4,460.0 |
| gopher | 2,321.0 | 8,777.4 | 70,354.6 | **120,500.9** | 10,022.8 | 582.0 | 1,200.0 |
| gravitar | **2,672.0** | 473.0 | 1,419.3 | **1,598.5** | 303.5 | 805.0 | 700.0 |
| hero | 25,763.0 | 20,437.8 | **55,887.4** | 31,655.9 | 32,464.1 | | 18,220.0 |
| ice_hockey | 0.9 | -1.9 | 1.1 | **33.0** | -2.8 | 4.1 | 2.0 |
| jamesbond | 406.7 | | | **21,322.5** | 541.0 | | 650.0 |
| kangaroo | 3,035.0 | 7,259.0 | **14,637.5** | 1,416.0 | 94.0 | 11,200.0 | 11,200.0 |
| krull | 2,395.0 | 8,422.3 | 8,741.5 | **11,741.4** | 5,560.0 | 8,647.2 | 10,889.0 |
| kung_fu_master | 22,736.0 | 26,059.0 | 52,181.0 | **97,829.5** | 28,819.0 | | 62,000.0 |
| montezuma_revenge | **4,367.0** | 0.0 | 384.0 | **2,500.0** | 67.0 | 0.0 | 0.0 |
| ms_pacman | **15,693.0** | 3,085.6 | 5,380.4 | **11,255.2** | 653.7 | | 3,410.0 |
| name_this_game | 4,076.0 | 8,207.8 | 13,136.0 | **25,783.3** | 10,476.1 | 4,503.0 | 7,210.0 |
| phoenix | | 8,485.2 | 108,528.6 | **224,491.1** | 52,894.1 | 4,041.0 | 3,810.0 |
| pitfall | | -286.1 | **0.0** | -0.6 | -78.5 | **0.0** | **0.0** |
| pong | 9.3 | 19.5 | 20.3 | 20.9 | 5.6 | **21.0** | -20.0 |
| private_eye | **69,571.0** | 146.7 | 4,234.0 | 49.8 | 206.9 | 100.0 | **15,200.0** |
| qbert | 13,455.0 | 13,117.3 | 33,817.5 | **302,391.3** | 15,148.8 | 147.5 | 5,125.0 |
| riverraid | 13,513.0 | | | **63,864.4** | 12,201.8 | 5,009.0 | 3,410.0 |
| road_runner | 7,845.0 | 39,544.0 | 62,041.0 | **222,234.5** | 34,216.0 | 16,590.0 | 15,900.0 |
| robotank | 11.9 | 63.9 | 61.4 | **73.8** | 32.8 | 11.9 | 16.0 |
| seaquest | 20,182.0 | 5,860.6 | 15,898.9 | **392,952.3** | 2,355.4 | 1,390.0 | 1,020.0 |
| skiing | | -13,062.3 | -12,957.8 | -10,789.9 | -10,911.1 | -15,442.5 | **-5,564.0** |
| solaris | | 3,482.8 | 3,560.3 | 2,892.9 | 1,956.0 | 2,090.0 | **7,200.0** |
| space_invaders | 1,652.0 | 1,692.3 | 18,789.0 | **54,681.0** | 15,730.5 | 678.5 | 840.0 |
| star_gunner | 10,250.0 | 54,282.0 | 127,029.0 | **434,342.5** | 138,218.0 | 1,470.0 | 800.0 |
| tennis | -8.9 | 12.2 | 0.0 | **23.9** | -6.3 | 4.5 | 0.0 |
| time_pilot | 5,925.0 | 4,870.0 | 12,926.0 | **87,085.0** | 12,679.0 | 4,970.0 | 16,800.0 |
| tutankham | 167.6 | 68.1 | 241.0 | **272.6** | 156.3 | 130.3 | 174.0 |
| up_n_down | 9,082.0 | | | **401,884.3** | 74,705.7 | 67,974.0 | 21,940.0 |
| venture | 1,188.0 | 163.0 | 5.5 | **1,813.0** | 23.0 | 760.0 | 1,400.0 |
| video_pinball | 17,298.0 | 196,760.4 | 533,936.5 | **565,163.2** | 331,628.1 | 22,834.8 | 36,157.0 |
| wizard_of_wor | 4,757.0 | 2,704.0 | 17,862.5 | **46,204.0** | 17,244.0 | 3,480.0 | 4,300.0 |
| yars_revenge | | 18,089.9 | 102,557.0 | **148,594.8** | 7,157.5 | 16,401.7 | 25,223.5 |
| zaxxon | 9,173.0 | 5,363.0 | 22,209.5 | **42,285.5** | 24,622.0 | 6,380.0 | 4,900.0 |
| State of the Art | 4 | 0 | 6 | **41** | 0 | 4 | 6 |

Table 5: **Extended Atari results.** For SOTA counts, we include ties as a point in that column (e.g. Rainbow, ES, and the GA get a point for Pitfall). Games for which scores are not reported in other papers are left blank and do not factor into head-to-head tallies.

| | Human | DQN | Rainbow | Ape-X | A3C | ES | GA 6B |
|---|---|---|---|---|---|---|---|
| Human | | 22, 24, 0 | 37, 9, 0 | 43, 6, 0 | 28, 21, 0 | 15, 28, 1 | 21, 28, 0 |
| DQN | | | 48, 4, 0 | 48, 4, 0 | 30, 22, 0 | 18, 29, 1 | 22, 29, 1 |
| Rainbow | | | | 43, 9, 0 | 11, 41, 0 | 7, 39, 2 | 11, 39, 2 |
| Ape-X | | | | | 3, 52, 0 | 7, 43, 0 | 7, 48, 0 |
| A3C | | | | | | 18, 32, 0 | 27, 28, 0 |
| ES | | | | | | | 28, 19, 3 |
| GA 6B | | | | | | | |

Table 6: **Head-to-head comparison between algorithms on Atari games.** Values represent the number of wins, losses, and ties between algorithms (e.g. vs. DQN, the GA wins on 22 games, loses on 29, and ties on 1. As discussed in the main text, apples-to-apples comparisons are difficult to make, as different algorithms exhibit different tradeoffs in computation, wall-clock speed, and data efficiency.

