# OpenReview forum: "Deep Neuroevolution: Genetic Algorithms are a Competitive Alternative for Training Deep Neural Networks for Reinforcement Learning"
_ICLR.cc/2019/Conference_

### Official Review · AnonReviewer2 · 2018-10-13
**Genetic algorithms are a potential alternative to backpropagation-based algorithms for reinforcement learning tasks**

**Rating:** 7
**Confidence:** 5

**Review:**

Post-rebuttal update: The review process has identified several issues such as missing citations and lack of clarity with respect to aims of the paper. Although the authors have failed to update the paper within the rebuttal period, their responses show an understanding of the issues that need to be addressed as well as a broad appreciation of work in EC that would be included in a final version, making it a useful resource for the wider ML community. On top of this they will include an even larger amount of empiricial data from the experiments they have already run, which is a valuable resource considering the amount of compute needed to obtain this data.

---

The current landscape of reinforcement learning - particularly in domains with high-dimensional structured input spaces such as images or text - relies heavily on backpropagation-based reinforcement learning algorithms.  An alternative that has re-emerged is ES, due to its simplicity and scalability. However, ES can also be considered a gradient-based method. In this paper, the authors apply a similar treatment to GAs, another simple method at its most basic. The authors claim to have 3 main contributions: extending the scale to which GAs can operate, suggesting that gradient-based methods may not achieve the best performance, and making available a vast array of techniques available from the EC literature; they demonstrate the latter by utilising novelty search (NS).

In my opinion the authors do indeed have a valuable contribution in a) demonstrating that a simple GA can successfully be applied to larger networks than was previously thought to be possible and b) introducing a novel software implementation that allows GAs to be efficiently scaled/distributed (similar in nature to the work of Salimans et al.). This is by itself valuable, as, along with recent work on ES, it potentially extends the range of problems that are perhaps best tackled using black-box optimisation techniques. Going against the prevailing trends in order to investigate alternative methods is an underappreciated service to the community, and I believe the evaluation of the methods and the choice of comparative methods to be just about satisfactory. As exemplified by NS, there is a wealth of techniques from the EC literature that could be applied to many topical problems, and the authors' main contributions opens up the road for this.

A lot of care has been put into evaluation on Atari games. The details in the main paper and supplementary material, with, e.g., clear definitions of "frames", make me believe that fair comparisons have taken place. All methods in the table, including GAs, perform best at some games (except for RS, which is a necessary baseline for GAs). It would be better to provide more data points that relate to prior works - such as scores at 200M frames to evaluate sample complexity (indeed, the authors note that good solutions can be found by GAs within a few generations, so it would be best to tabulate this) and at ~ 4d to evaluate wall-clock time (is it possible to push performance even further?). Since the GA presented is very rudimentary, I consider the baselines in the main paper to be reasonable, but it would be misleading to not present newer work. The authors do so in the supplementary material, and it is promising to note that the GA still achieves state-of-the-art performance in a few games even when compared to the most sophisticated/distributed state-of-the-art DRL algorithms developed in a concentrated effort over the last few years. Despite having an NS variant, it is a shame that the authors did not show that this could potentially improve performance on Atari, when BCs such as the game RAM or preferably random CNN features are easily available.

The authors also evaluate on a maze that is a staple task in the EC literature to demonstrate the power of NS. While the results are unsurprising, it is a reasonable sanity check. The final evaluation is on a difficult continuous control task, in which GAs solve the task, but have much poorer sample complexity than ES. Given the range of continuous control tasks used to benchmark RL algorithms nowadays, the authors would do well to present results across more of these tasks. Again, NS was not evaluated on this task.

A major weakness of this paper is the presentation. The authors discuss some interesting findings, but would be better served by being more concise and focused. In particular, the emphasis should be more on showcasing quantitative results. Doing so, with more continuous control tasks, would make the claims of this paper more substantiated.

---

> ### Author Response · Authors · 2018-11-27
> **Response to Reviewer 2**
>
> Thank you for you helpful review. We are glad that you identify our paper as a meaningful contribution and agree with you about the underappreciation of going against prevailing trends.
>
> We agree that there are many dimensions one can compare algorithms (e.g. sample complexity, wall-clock time, computation complexity) and have made an effort to release checkpoint data as well as training data when we open source the code. Unfortunately, due to limited space, we couldn’t be exhaustive with those comparisons in the main text, but we will add more results on our experiments to the SI, namely results on Atari after a handful of generations as well as 200M and 1B frames for all Atari games. It is important to note that the optimal hyperparameters change depending on the comparison being made and it can be misleading to compare algorithms across many dimensions without appropriate tuning. For example, there is a tradeoff between sample efficiency and scalability when selecting a population size as well as the selection pressure. We will also add plots of GA performance on more continuous control tasks (other Mujoco robot control domains).
>
> We also believe that extending the GA+Novelty Search to Atari and MuJoCo tasks is an important future direction, but we think that a thorough analysis of GA+NS on high-dimensional problems requires and deserves its own publication.

---

> > ### Comment · AnonReviewer2 · 2018-11-27
> > **Promised Changes?**
> >
> > Your proposed revisions in response to all of the reviewers' comments sound promising, but there doesn't seem to be any updates to the paper within the (extended) rebuttal period?

---

> > > ### Author Response · Authors · 2018-11-28
> > > **Response to Reviewer 2**
> > >
> > > We are glad you are happy with the proposed changes, or at least find them promising. We thought the effect of most of them (such as adding citations and columns to tables) was predictable such that we would happily make them if the paper is accepted. It sounds instead like you would like to see them first to make your decision. We apologize for not anticipating that. Unfortunately the rules do not allow us to upload a changed PDF at this point.
> > > However, if there are any text additions you would like to review we are allowed to add them here in our reply. Please let us know if there is any newly proposed text you would like to review. Thanks!

---

### Official Review · AnonReviewer1 · 2018-10-30
**An empirical evaluation of a specific genetic algorithm, but without much value add**

**Rating:** 3
**Confidence:** 4

**Review:**

This paper looks at a specific implementation of a "genetic algorithm" (GA) when applied to learning Atari games.
Using a black-box "random search" technique, with a few heuristic adaptations, they find performance that is roughly competitive with several other baseline methods for "Deep RL".
More generally, the authors suggests that we should revisit "old" algorithms in Machine Learning and that, when we couple them with larger amounts of computation, their performance may be good.

There are several things to like about this paper:
- The authors place a high importance on implementation details + promise to share code. This seems to be a paper that is heavily grounded in the engineering, and I have high confidence the results can be reproduced.
- The algorithm appears broadly competitive on several Atari games (although Table 1 is admittedly hard to parse).
- The algorithm is generally simple, and it's good to raise questions of baseline / what are we really accomplishing.

However, there are several places where this paper falls down:
- The writing/introduction is extremely loose... terms are used and introduced without proper definition for many pages.
    + How would be think about the venn diagram of "evolutionary strategies", "genetic algorithms", "deep Q networks", "deep RL algorithms" and "random search"... there is clearly a lot of overlap here.
    + The proposed deep GA has a "deep Q network" (or is it a policy... the paper does not make this clear), forms a type of "evolutionary strategy" and, at its heart is a type of "random search", but is it not also a "deep RL algorithm"?
    + It is not until page 3 that we get a proper definition of the algorithm, and it's hard to keep track of the differences the authors want to highlight compared to the "baselines".
    + What do we gain from the claim "old algorithms work well"... gradient descent is also an old algorithm, as are seemingly all of the alternatives? Is age in-of-itself an asset?
    + Statements like "compression rate depends on the number of generations, but in practice is always substantial" are very loose... what does the "practice" refer to here, and why?

- There is very little insight/analysis into *how* or *why* this algorithm performs better/worse than the alternatives. I don't feel I understand if/when I should use this algorithm versus another apart from a wall of seemingly random Atari results. In fact, there is a large literature that explains why GAs give up a lot of efficiency due to their "black box" nature... what do the authors think of this?

- This paper seems purely focused on results rather than insight, with many hacks/tweaks to get good results... should we believe that GA+novelty search is a general algorithm for AI, or is it just another tool in the arsenal of a research engineer?
    + In the end, the algorithm doesn't actually seem to outperform state-of-the-art on these Atari baselines... so what are we supposed to take away from this.

Overall, I don't think that this paper provides very much in the way of scientific insight.
Further, the results are not even superior to existing algorithms with stronger groundings.
For me, this leads it to be a clear reject... even though they probably have code that reliably solved Atari games in a few hours.

---

> ### Author Response · Authors · 2018-11-27
> **Response to reviewer 1**
>
> Thank you for you helpful review. We are glad you like the simplicity of the algorithm. You are right that we spent a lot of engineering and scientific effort to create reproducible results and follow evaluating procedures from the RL community, and we are glad that effort did not go unnoticed. We are also happy you too find the results surprising. Most people assumed GAs would fail at this scale and on this challenging a task. Surprisingly, vanilla GAs perform roughly as well as the original DQN paper, a performance level that merited a Nature cover and launched an avalanche of interest/research in Deep RL. We believe this result is interesting and will motivate future research and thus deserves to be published.
>
> We will clarify/tighten the language. While GAs can solve RL tasks, they are not RL algorithms (at least not as defined by Sutton & Barto 1998 [1]). We will clarify the distinctions and similarities between GAs, ES, and RL algorithms in our final manuscript.
>
> Our paper shows that GAs outperform traditional deep RL algorithms on some domains, yet understanding in which situations GAs are most preferable is a promising direction for follow-up work. Although the answers are currently unclear, we believe that motivating such questions is a helpful service provided by the paper. Providing definitive answers about when to use GAs will require its own follow-up research, but we will add a discussion of our own hypotheses to this paper.
> Briefly, in some of the games, we identified that clipping reward between [-1,1], a trick commonly used in deep RL, was preventing other algorithms from performing well (e.g. Centipede and Bowling). In others (e.g. Solaris, PrivateEye, and Frostbite), we think the mutation-based exploration is more effective at finding hidden/tricky high-rewards. We also found some games that are significantly harder from a black-box optimization perspective. In the game Enduro for example, a reward of +1 is given for passing a car, and a reward of -1 is given for being passed. While an RL algorithm like DQN and A3C would learn from both events, a black-box optimizer like ES and GA would only learn from the final difference between cars passed and cars passed by.
>
> Aside from their competitive performance, another set of reasons one might prefer GAs (as we mention) is that they are more amenable to parallelization and have faster wall-clock speed than most other algorithms. They are even more effective when a large number of parallel workers can be instantiated. Thus our simple GA is most effective in scenarios where parallel compute is available and wall-clock speed is paramount. Overall, however, we think that determining when each method performs better or worse is a hard thing to tease apart and the current state of RL literature is a great example (e.g. when should you use DQN vs. PPO vs. A3C vs. TRPO?).
>
> We understand and agree with the statement made about sample efficiency of black-box algorithms, but sample efficiency is just one dimension of the quality of an algorithm. In reality, many applications of RL, specifically those for which good simulators are available, are not concerned with sample efficiency, but with computation efficiency, time, or even just final performance. Furthermore, it is interesting to show that gradient-based methods are not the only way of solving these tasks since many useful techniques are non-trivial to implement with gradient-descent (e.g. architecture search and networks with limited-precision weights).
>
> We will clarify in the manuscript that although we are using the same architecture used for DQN and A3C, our method optimizes a deterministic policy. Actions are taken based on an argmax over the output of the network, which do not represent specific quantities, such as Q-values.
>
> [1] Richard S Sutton and Andrew G Barto. Reinforcement learning: An introduction, volume 1. MIT press Cambridge, 1998.

---

> > ### Comment · AnonReviewer1 · 2018-11-27
> > **Answering response**
> >
> > - DQN paper was particularly impactful as *first* method to solve Atari from pixels with neural network. You are right that this method achieves similar results, but I don't think that the observation: "you can swap SGD for GA" is anywhere near the same scale.
> >
> > - I'm not sure why GAs would not be RL algorithms according to Sutton & Barto? Reading the first paragraphs of their second edition it certainly seems like it would be a valid RL algorithm?
> >
> > - I think it's important to dig deeper into these questions/understanding. This response is a good start, maybe you can distill these insights to clear experiments?
> >
> > - Again, benefits to parallelization would be good to distill into clear insights. It seems that many gradient-style algorithms also do very well with large scale parallelization? e.g. BigGAN, IMPALA... etc
> >
> > - It's very important to disambiguate two common types of "RL" research.
> > To quote John Langford https://opendatascience.com/pervasive-simulator-misuse-with-reinforcement-learning/
> > """
> > The surge of interest in reinforcement learning is great fun, but I often see confused choices in applying RL algorithms to solve problems. There are two purposes for which you might use a world simulator in reinforcement learning:
> > 1) Reinforcement Learning Research: You might be interested in creating reinforcement learning algorithms for the real world and use the simulator as a cheap alternative to actual real-world application.
> > 2) Problem Solving: You want to find a good policy solving a problem for which you have a good simulator.
> > In the first instance I have no problem, but in the second instance, I’m seeing many head-scratcher choices.
> > """
> > So, if we want to compare GA vs simulator-based planners (2) then I think it's very important to compare to other methods like MCTS.
> >
> > Overall, I thank the authors for their response, but choose to keep my score unchanged.

---

### Official Review · AnonReviewer6 · 2018-11-13
**Interesting exploration, but lacks needed rigor.**

**Rating:** 4
**Confidence:** 4

**Review:**

This paper demonstrates that Genetic Algorithms can be used to train deep neural policies for Atari, locomotion, and an image-based maze task. It's interesting that GAs can operate in the very high-dimensional parameter space of DNNs. Results show that on the set of Atari games, GAs perform roughly as well as other ES/DeepRL algorithms.

In general, it's a bit hard to tell what the contribution of this paper is - as an emperical study of GA's applied to RL problems, the results raise questions:

1) Why only 13 Atari games? Since the algorithm only takes 1-4 hours to run it should only take a few days to collect results on all 57 games?

2) Why not examine a standard GA which includes the crossover operator? Do the results change with crossover?

3) The authors miss relevant related work such as "A Neuroevolution Approach to General Atari Game Playing" by Hausknecht et al., which examines how neuroevolution (GA based optimization which modifies network topology in addition to parameters) can be used to learn policies for Atari, also scaling to million-parameter networks. This work already showed that GA-based optimization is highly applicable to Atari games.

4) Is there actual neuroevolution going on? The title seems to imply there so, but from my reading of the paper - it seems to be a straightforward GA (minus crossover) being applied to weight values without changes to network topology.

I think this paper could be strengthened by providing more insight into 1) Given that it's already been shown that random search can be competitive to RL in several Mujoco tasks (see "Simple random search provides a competitive approach to reinforcement learning") I think it's important to understand why and in what scenarios GAs are preferable to RL and to ES given similar performance between the various methods. 2) Analysis as to whether Atari games in particular are amenable to gradient-free optimization or if GA's are equally applicable to the full range or RL environments?

---

> ### Author Response · Authors · 2018-11-27
> **Response to Reviewer 6**
>
> Thank you for you helpful review, we are glad you think are findings are interesting.
>
> We did compare on all 57 games and provide head-to-head comparisons for all algorithms on them. Those results were summarized in the main text, and reported SI Table 5. The table in the main paper summarizes results for 13 games, which we thought was more digestible. We can move all 57 to the main text if you prefer.
>
> We realize that previous work has looked at GAs for playing Atari games, and will add the relevant citations to our paper. Specifically we will add a comparison to Hausknecht et al (HEA)[1] which seems to be the most relevant work solving Atari games with GAs. We used the same preprocessing, architecture, and evaluation routine as DQN (A3C and ES follow similar protocols) so that the algorithms can be compared. All three of those are significantly different in HEA, which makes algorithmic comparisons difficult. The most important difference is the evaluation scheme. HEA report the max score found for one set of initial game conditions, allowing memorization of an open-loop sequence of instructions that perform well on that one specific initial condition. They did not test whether these policies generalize to other initial conditions. We report average scores over many different initial conditions. With their evaluation method, the GA outperforms the HEA results on 50 of the 55 games and ties on 2. Additionally, HEA has a different, much smaller input representation (an 8-dimensional one-hot SECAM color scheme and down-sampling to 16x21 instead of 84x84). Finally, their architecture has 4x fewer parameters, a single hidden layer only, and is indirectly encoded. We will highlight the important historical contribution of the HEA paper and describe these differences in the revision.
>
> The term “neuroevolution” is broad and covers any type of using evolutionary algorithms to optimize neural network, including evolving the weights only or evolving the weights and topology. We also wanted to provide a fair comparison with other deep RL algorithms, which are not allowed to change their architectures during optimization. We wanted to evaluate the simplest possible GA and found that the GA was competitive without crossover so we did not try adding that technique. Crossover, as well as many other evolutionary techniques, would likely further improve the performance of the GA.
>
> Our paper shows that GAs outperform traditional deep RL algorithms on some domains, yet understanding in which situations GAs are most preferable is a promising direction for follow-up work. Although the answers are currently unclear, we believe that motivating such questions is a helpful service provided by the paper. Providing definitive answers about when to use GAs will require its own follow-up research, but we will add a discussion of our own hypotheses to this paper.
> Briefly, in some of the games, we identified that clipping reward between [-1,1], a trick commonly used in deep RL, was preventing other algorithms from performing well (e.g. Centipede and Bowling). In others (e.g. Solaris, PrivateEye, and Frostbite), we think the mutation-based exploration is more effective at finding hidden/tricky high-rewards. We also found some games that are significantly harder from a black-box optimization perspective. In the game Enduro for example, a reward of +1 is given for passing a car, and a reward of -1 is given for being passed. While an RL algorithm like DQN and A3C would learn from both events, a black-box optimizer like ES and GA would only learn from the final difference between cars passed and cars passed by.
>
> Aside from their competitive performance, another set of reasons one might prefer GAs (as we mention) is that they are more amenable to parallelization and have faster wall-clock speed than most other algorithms. They are even more effective when a large number of parallel workers can be instantiated. Thus our simple GA is most effective in scenarios where parallel compute is available and wall-clock speed is paramount. Overall, however, we think that determining when each method performs better or worse is a hard thing to tease apart and the current state of RL literature is a great example (e.g. when should you use DQN vs. PPO vs. A3C vs. TRPO?).
>
>
> Finally, we note that the “Simple Random Search” paper mentioned is significantly different from what we described as random search. Their method is a random search for the gradient and is in practice very similar to OpenAI’s ES algorithm.
>
> [1] M Hausknecht, J Lehman, R Miikkulainen, P Stone. A Neuroevolution Approach to General Atari Game Playing. IEEE Transactions on Computational Intelligence and AI in Games, DOI:10.1109/TCIAIG.2013.2294713:1–18, 2013

---

### Official Review · AnonReviewer3 · 2018-11-13
**Explore GAs as alternative for DeepRL**

**Rating:** 6
**Confidence:** 2

**Review:**

The authors explore the use of GAs as an alternative to gradient based methods for DeepRL and show performance comparisons indicating competitive /on par performance when compared to the existing methods.

Pros:
-I liked the idea of exploring other avenues for approaching DeepRL problems, and challenging existing paradigms or trends.
This has a couple of implications - it might lead to expanding the toolbox for DeepRL problems, and it also might lend insight regarding these problems that could lead to new directions in exploring other methods. In other words, exploring WHY GA do better (and even RS!) could be useful, and this study is a foray in that direction.
It could also (as they point out) lead to useful hybrid approaches.
-Their implementation will be made available and includes some useful features (compact network encoding, etc).
-They demonstrate very nice speeds that can be impactful especially for more resource limited scenarios.

Cons:
-There has been some exploration of use of GA for games, though not in DeepRL (AFAIK). They should cite previous efforts and explain what is their contribution specifically above and beyond previous efforts.
-Although I applaud the authors for making exploratory efforts into this alternative approach and demonstrating competitive performance, the reader is left with some degree of "so what?". In other words, WHEN should this method be applied? How does it fit into the existing toolkit?
-This is in some ways a conceptual exploration, and it shows interesting things that open "why" questions. Why does GA do better in some cases? They address this nicely in the discussion but more exploration in that direction would be useful.

Overall I like this paper for its conceptual contribution that may provide insight that will help the field grow in interesting directions, and for the implementation it will provide.

---

> ### Author Response · Authors · 2018-11-27
> **Response to Reviewer 3**
>
> Thank you for you helpful review. We are glad you think our work explores new avenues for approaching deep RL and opens up interesting possibilities for future work.
>
> We realize that previous work has looked at GAs for playing Atari games, and will add the relevant citations to our paper. For example, as another reviewer requested, we’ll cite and compare to Hausknecht et al (HEA)[1]. Briefly, in contrast to that work, our work operates on much higher-dimensional input, with a much deeper policy, and on a stochastic version of each game, rather than a deterministic version for which the policy can be memorized.
>
> Our paper shows that GAs outperform traditional deep RL algorithms on some domains, yet understanding in which situations GAs are most preferable is a promising direction for follow-up work. Although the answers are currently unclear, we believe that motivating such questions is a helpful service provided by the paper. Providing definitive answers about when to use GAs will require its own follow-up research, but we will add a discussion of our own hypotheses to this paper.
> Briefly, in some of the games, we identified that clipping reward between [-1,1], a trick commonly used in deep RL, was preventing other algorithms from performing well (e.g. Centipede and Bowling). In others (e.g. Solaris, PrivateEye, and Frostbite), we think the mutation-based exploration is more effective at finding hidden/tricky high-rewards. We also found some games that are significantly harder from a black-box optimization perspective. In the game Enduro for example, a reward of +1 is given for passing a car, and a reward of -1 is given for being passed. While an RL algorithm like DQN and A3C would learn from both events, a black-box optimizer like ES and GA would only learn from the final difference between cars passed and cars passed by.
>
> Aside from their competitive performance, another set of reasons one might prefer GAs (as we mention) is that they are more amenable to parallelization and have faster wall-clock speed than most other algorithms. They are even more effective when a large number of parallel workers can be instantiated. Thus our simple GA is most effective in scenarios where parallel compute is available and wall-clock speed is paramount. Overall, however, we think that determining when each method performs better or worse is a hard thing to tease apart and the current state of RL literature is a great example (e.g. when should you use DQN vs. PPO vs. A3C vs. TRPO?).
>
> [1] M Hausknecht, J Lehman, R Miikkulainen, P Stone. A Neuroevolution Approach to General Atari Game Playing. IEEE Transactions on Computational Intelligence and AI in Games, DOI:10.1109/TCIAIG.2013.2294713:1–18, 2013

---

### Official Review · AnonReviewer5 · 2018-11-13
**GAs for Deep-RL; good performance numbers, but needs more intuition and understanding.**

**Rating:** 6
**Confidence:** 4

**Review:**

The authors show that using a simple genetic algorithm to optimize the weights of a large DNN parameterizing the action-value function can result in competitive policies. The GA uses selection truncation and elitism as heuristics; compact-encoding of the genome to improve system efficiency. Results on Atari are presented, along with a comparison with standard gradient-based RL algorithms and ES.

Pros:

The knowledge that on a moderately complex optimization problem such as RL on Atari with pixel inputs, learning can be achieved using parameter perturbation (combined with heuristics) is valuable to the community. The authors’ motivation --- that GAs could prove to be another interesting tool for RL --- is well-founded. The efficient implementation using CPU-GPU hybridization and the compact-encoding is a good engineering contribution and could help accelerate further research.

Cons:

a.) One issue is that the results are presented in a somewhat hard-to-read manner. Table 1. shows numbers on 13 selected Atari games, of which 3 work best with GA (considering 1B frames). Comparison on all games helps to understand the general applicability of GAs. This is provided in Table 6., but the GA (1B) is missing. So are we comparing ES (1B) vs GA (6B) in Table 6 on all games? I can understand that DQN/A3C take days to run that many frames, but what does a ES (1B) vs GA (6B) comparison tell us? Am I reading it right?

b.) Deep RL algorithms are approximations of theoretically sound ideas; so is ES in a way. What would help make this an even better paper is if the authors attempt to demystify the GA results. For example, I would have enjoyed some discussion on the potential reasons for the superiority of GAs on the 3 games in Table 1. Are these hard exploration environments and the parameter-space exploration in GA outperforms the action-space (e-greedy) exploration in A3C (DQN)? Random search is also better than DQN on those 3 environments --- Is the availably of good policies around the initial weight distribution contributing to the GA results in anyway? Furthermore, it would be interesting to pick out the worst games for GA from Table 5. (cases of catastrophic failure of GA), and analyze the shortcomings of GA. This would be helpful in further advancing research in GAs for deep RL. As a suggestion, ablation studies could go some distance in improving the interpretability, since the GA in the paper uses critical choices (parameter-noise variance, population-size etc.) and heuristics (truncation, elitism, etc.) that affect performance w.r.t baselines.

Other comments:

Comparing GA+NS with A3C/DQN under environments with deceptive rewards is not completely fair. If the GA is armed with an exploration strategy like NS, so should the standard RL algorithms (e.g. by adding count-based, curiosity exploration).

Section 4.3 (Humanoid discussion) could be move to Appendix since it currently doesn’t work with GA, and if anything, increases doubts on the robustness of GA since the Humanoid DNN is fairly small.

---

> ### Author Response · Authors · 2018-11-27
> **Response to Reviewer 5**
>
> Thank you for you helpful review. We are glad you think our results are valuable to the community and that our motivation is well-funded.
>
> We were not able to fit both GA (1B) and GA (6B) on Table 6 and decided to go with the 6B version as a middle ground comparison considering all algorithms. Having ES (6B) would have been a great addition, but unfortunately those results are not published and the Atari ES code and hyperparameters were not open-sourced. While we agree we shouldn’t compare GA (6B) with ES (1B), we thought reimplementing ES and running to 6B frames without proper hyperparameters would be misleading and unfair to it. To offer a more fair comparison to ES we will add GA (1B) to Table 6. Against ES-1B, the GA-1B won on 23 games and lost on 25.
>
> In an ideal world we would be able to try all possible GAs, or at least many variants. But to even do the suggested ablations could cost over $10,000 in computation (based on EC2 pricing). The experiments we include in the paper already involved a massive amount of computational resources, even after our engineering advances to accelerate it. That is partly because to improve statistical quality, we ran our algorithm multiple times for each game which costs dozens of GPU days. Thus, doing a sweep of GA variants is beyond our resources unfortunately. Additionally, our main point was that a simple GA works, not that this is the best GA.
>
> We will clarify in the manuscript that although we are using the same architecture used for DQN and A3C, our method optimizes a deterministic policy. Actions are taken based on an argmax over the output of the network, which do not represent specific quantities, such as Q-values.
>
> We agree (and say in the paper) that there are exploration algorithms for traditional RL algorithms. By comparing GA+NS on the hard maze with A3C/DQN we simply meant to show that on a problem in which exploration is necessary (as shown by the fact that vanilla A3C/DQN do not solve the problem), there is an off-the-shelf GA enhancement that works to encourage exploration. We wanted to show one such example of an extension to the GA that we could test in a high-dimensional task. We will make it even clearer that we believe intrinsic motivation algorithms for RL would solve the hard maze task too.
>
> We will add a discussion about our hypotheses for why the GA (or RS) performs well or poorly on some games. In some of the games, we identified that clipping reward between [-1,1], a trick commonly used in deep RL, was preventing other algorithms from performing well. In others (e.g. Solaris, PrivateEye, Frostbite), we think the mutation-based exploration is more effective at finding hidden/tricky high-rewards. We also found some games that are significantly harder from a black-box optimization perspective. In the game Enduro for example, a reward of +1 is given for passing a car, and a reward of -1 is given for being passed. While an RL algorithm like DQN and A3C would learn from both events, a black-box optimizer like ES and GA would only learn from the final difference between cars passed and cars passed by.
>
> Regarding theoretical guarantees: They are less well known in the RL community, but much theoretical work including some theoretical guarantees also exist for genetic algorithms (e.g. [1]). We will summarize them and add cites to the paper. In general, the gap between theory and practice is well known, whether in traditional RL or evolutionary algorithms. While the main contribution of our work was to provide empirical evidence that GAs can achieve competitive results in high-dimensional problems, we hope it will also spur research into the theoretical properties of GAs in high-dimensional spaces -- similar to how the empirical success of SGD-based deep learning has driven theoretical results and deeper understanding.
>
> [1] Lothar M. Schmitt. Theory of genetic algorithms. Theoretical Computer Science, Volume 259, Issues 1–2, 2001, Pages 1-61.

---

> > ### Comment · AnonReviewer5 · 2018-11-29
> > **Post-rebuttal update**
> >
> > The unclarity about the why/when/how of "GAs for deep RL" persists. However, after further consideration, I believe there's reason to be optimistic about this direction of research ("optimism in the face of uncertainty"; no pun intended). Different from what one might infer by reading the paper title, GAs currently don't seem to be a competitive alternative to (SOTA) RL (7-48 compared to Ape-X on Atari). But as mentioned in the manuscript, research on optimizing the efficiency and robustness of GAs may get us there. I'm raising my score by 1 point - (6): Marginally above acceptance threshold

---

> > > ### Author Response · Authors · 2018-12-02
> > > **Thank you**
> > >
> > > Thank you. We greatly appreciate your open-minded consideration and that you raised your score for our paper.

---

### Public Comment · (anonymous) · 2018-10-02
**About the experiments.**

In your implementation,  do you use the OpenAI gym Atari  2600 lib? If not, how much performance improvement can you get by using your own game environment compared with using the OpenAI gym? Thank you very much.

---

> ### Author Response · Authors · 2018-10-03
> **Re: About the experiments.**
>
> Thank you for your interest in our paper. We use a modified version of OpenAI's atari-py+gym library. It is functionally identical, but improves throughput by up to 200% depending on CPU/GPU usage and the number of cores. We modified atari-py to make the constructors thread-safe (no speedup). We replaced OpenAI/gym environments with custom Tensorflow operations. Again, these changes do not change the domain functionally, but provide speedups. We also made use of batched Tensorflow operations for the preprocessing steps. We will make these details clear in an updated supplementary info section in the paper. We will also open-source the code with the paper.

---

### Meta-Review · Area_Chair1 · 2018-12-13
**Improvement needed.**

**Confidence:** 5
**Recommendation:** Reject

**Metareview:**

This paper presents an empirical study of the applicability of genetic algorithms to deep RL problems. Major concerns of the paper include: 1. paper organization, especially the presentation of the results, is hard to follow; 2. the results are not strong enough to support that claims made in this paper, as GAs are currently not strong enough when compared to the SOTA RL algorithms; 3. Not quite clear why or when GAs are better than RL or ES; Lack of insights. Overall, this paper cannot be accepted yet.